# Unitary Convolutions for Message-passing and Positional Encodings on Directed Graphs

## Abstract

In many real-world networks, relationships are inherently directional, yet most graph neural networks (GNNs) assume undirected edges, and naïve adaptations of undirected GNNs to directed graphs amplify oversmoothing and gradient pathologies that cap model depth. Unitary graph convolutions (UniConv) provably prevent representational collapse and oversmoothing, but cannot incorporate edge directionality or edge features. In this paper, we introduce a **d**irected **un**itary GNN with **e**dge features (**Dune**), which retains these guarantees while overcoming UniConv's limitations by incorporating edge directionality and edge features. Dune keeps gradient norms bounded at any number of layers, allowing it to benefit from neural network depth, unlike existing directed GNNs. The same unitary operator can be embedded in hybrid architectures with graph transformers, where its wavelike propagation supplies positional information and reduces the importance of random-walk or Laplacian-based encodings. We prove that Dune avoids exponential oversmoothing that plagues existing directed GNNs, and empirically show that it achieves state-of-the-art performance on 12 directed-graph benchmarks while remaining trainable beyond 100 layers, improving performance by up to 18 percentage points over strong baselines. These results establish unitary convolutions as a scalable, geometry-aware foundation for deep learning on directed graphs. We make a preliminary version of our codebase available here.

## 1 Introduction

Machine learning methods for graph-structured data have achieved remarkable success across various domains, yet most existing graph neural networks (GNNs) assume undirected edges and symmetric information propagation. In numerous real-world applications—including communication networks, transportation systems, citation graphs, and electrical circuits—relationships between entities are inherently directional (Easley & Kleinberg, 2010; Newman, 2010). Ignoring such directionality discards essential information about signal, influence, or resource flows within networks, and naïve adaptations of undirected architectures typically exacerbate known issues rather than resolve them (Rossi et al., 2024).

Extending GNNs to directed graphs remains challenging. Common strategies include treating each directed edge as two opposing undirected edges, but this significantly increases computational costs without effectively differentiating source from target nodes during message passing (Tong et al., 2020). Other methods incorporate minor modifications to convolutional or attention mechanisms, often resulting in ad hoc solutions that neither ensure stable training nor resolve fundamental architectural issues (Jiang et al., 2025). More recently, transformer-based approaches have emerged, explicitly designed to capture directional information through positional encodings. These positional encodings are usually precomputed quantities such as random walk transition probabilities or Laplacian eigenvectors that are meant to capture the geometry of a directed graph (Geisler et al., 2023; Huang et al., 2024b).

However, despite the promise of transformer-based architectures with advanced positional encodings, existing approaches still face significant hurdles. Common positional encodings require careful hyperparameter tuning and - as in the undirected case - scale poorly with the size of the graph (Dwivedi & Rampášek, 2021; Kreuzer et al., 2021). The flexibility and scalability of these models is therefore limited. Most existing message-passing based GNNs (MPNNs) for directed graphs similarly

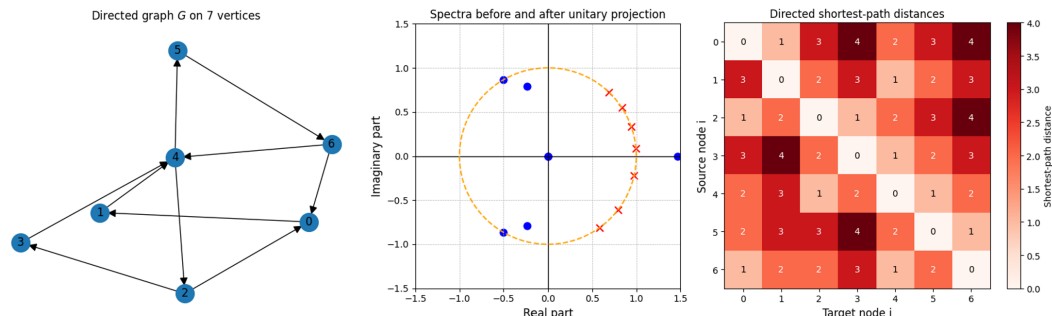

Figure 1: An example directed graph $G$ (left), the spectrum of its adjacency matrix before (blue) and after the mapping into the unitary group (red) that Dune employs (center), and the directed shortest distances between the nodes in $G$ (right). Dune excels at capturing these topological features.

suffer from architectural shortcomings, such as gradient instabilities or oversmoothing, where node representations become indistinguishable after several layers (Li et al., 2018; Alon & Yahav, 2021). Consequently, even state-of-the-art directed MPNNs often plateau in performance after relatively few layers, and cannot benefit from modeling complex, long-range dependencies (Cai & Wang, 2020).

In this paper, we introduce a directed unitary GNN with edge features (Dune), which builds on the recently proposed Unitary Graph Convolutional Neural Network (UniGCN) (Kiani et al., 2024). Dune extends UniGCN by incorporating edge directionality and rich edge features. By leveraging unitary transformations, Dune provably avoids oversmoothing and ensures gradient norms remain bounded regardless of network depth. Dune can also be shown to capture rich geometric information about an input graph, which can reduce the importance of positional encodings when using transformers on directed graphs. Empirically, these properties allow Dune to set a new state of the art for tasks on directed graphs.

**Contributions.** Our key contributions in this paper are as follows:

1. We propose Dune, a powerful unitary graph neural network that supports directed graphs and graphs with edge features.

2. We provide a theoretical analysis of oversmoothing on directed graphs and show that Dune provably avoids this. Vanishing and exploding gradients can similarly be ruled out with Dune.

3. We introduce the use of unitary convolutions in hybrid models, i.e. with graph transformers. We empirically and theoretically show that unitary convolutions encode rich geometric information that can reduce the importance of positional encodings.

4. Finally, we present experimental results on 12 directed graph datasets that show that Dune significantly outperforms the state of the art when used for message-passing or with graph transformers. Ablation studies show that unlike existing approaches, our model is still trainable with more than 100 layers.

| Model | Directionality | Edge Features | Expressivity | No Oversmoothing | PE Connection |
|---|---|---|---|---|---|
| Di-GCN | ✓ | ✗ | ✗ | ✗ | ✗ |
| MagNet | ✓ | ✗ | ✗ | ✗ | ✗ |
| Dir-GNN | ✓ | ✓ | ✓ | ✗ | ✗ |
| NDDGNN | ✓ | ✗ | ✗ | ✗ | ✗ |
| UniGCN | ✗ | ✗ | ✗ | ✓ | ✗ |
| **Dune (Ours)** | ✓ | ✓ | ✓ | ✓ | ✓ |

Table 1: Comparison of (directed) GNN models

## 2 PRELIMINARIES

We use $c, w$, and $M$ to denote scalars, vectors, and matrices, respectively. For a matrix $M$, its transpose and conjugate transpose are written $M^\top$ and $M^\dagger$. For matrices $A$ and $B$, their tensor (Kronecker) product is $A \otimes B$. For a vector $w$, the Euclidean norm is $\|w\|$, while for a matrix $M$, we use $\|M\|$ for the operator norm and $\|M\|_F$ for the Frobenius norm.

**Graph Neural Networks**   We write graphs as $\mathcal{G} = (V, E)$ with $V$ and $E$ denoting the sets of nodes and edges. For a graph on $n$ nodes, unless stated otherwise, we set $V = \{1, \ldots, n\}$ and denote its adjacency matrix by $A \in \mathbb{R}^{n \times n}$ and node features by $X \in \mathbb{R}^{n \times d}$. The diagonal (undirected) degree matrix $D \in \mathbb{R}^{n \times n}$ has entries $D_{ii}$ equal to the degree of node $i$, and the normalized adjacency is $\widetilde{A} = D^{-1/2} A D^{-1/2}$. On directed graphs, we denote the in-degree and out-degree matrices by $D_{\text{in}}$ and $D_{\text{out}}$, respectively. Given node features $X \in \mathbb{R}^{n \times d_{\text{in}}}$, where row $i$ stores the $d_{\text{in}}$-dimensional vector of node $i$, graph convolutions on undirected graphs take the general form (Kipf & Welling, 2017; Gilmer et al., 2017)

$$f_{\text{conv}}(X; A) = X W_0 + A X W_1 + \cdots + A^k X W_k, \tag{1}$$

with trainable parameters $W_0, \ldots, W_k \in \mathbb{R}^{d_{\text{in}} \times d_{\text{out}}}$. For simplicity, we often omit $A$ as an explicit argument. In practice, one usually includes a single "message passing" step, yielding $f_{\text{conv}}(X) = A X W$. Graph convolutions are permutation equivariant, since for any permutation matrix $P_\pi \in \mathbb{R}^{n \times n}$,

$$f_{\text{conv}}(P_\pi X; P_\pi A P_\pi^{-1}) = P_\pi f_{\text{conv}}(X; A). \tag{2}$$

**Group Theory Basics**   Symmetries ("invariances") are transformations that leave data properties unchanged and thus capture the geometric structure of the domain. Algebraically, they are modeled by groups. A group is a matrix Lie group if it is a closed subgroup of the set $GL(n, \mathbb{R})$ of invertible $n \times n$ matrices (Stillwell, 2008). Each Lie group has an associated Lie algebra, the tangent space at the identity. For an introduction, see (Hall, 2015). In this work we encounter the orthogonal $O(n)$ and unitary $U(n)$ groups,

$$O(n) = \{U \in \mathbb{R}^{n \times n} : UU^\top = I\}, \qquad U(n) = \{U \in \mathbb{C}^{n \times n} : UU^\dagger = I\}. \tag{3}$$

Their Lie algebras consist of skew-symmetric and skew-Hermitian matrices,

$$\mathfrak{o}(n) = \{M \in \mathbb{R}^{n \times n} : M + M^\top = 0\}, \qquad \mathfrak{u}(n) = \{M \in \mathbb{C}^{n \times n} : M + M^\dagger = 0\}. \tag{4}$$

For $M \in \mathfrak{o}(n)$ (or $\mathfrak{u}(n)$), the matrix exponential maps it into the group, i.e., $\exp(M) \in O(n)$ (or $U(n)$). (Kiani et al., 2024) use the exponential map to define a graph convolutional operator on undirected graphs as in equation 1 that is also unitary:

**Definition 1.** (Separable unitary graph convolution (UniConv)). Given an undirected graph $G$ over $n$ nodes with adjacency matrix $A \in \mathbb{R}^{n \times n}$, separable unitary graph convolution (UniConv) $f_{\text{Uconv}} : \mathbb{C}^{n \times d} \to \mathbb{C}^{n \times d}$ takes the form

$$f_{\text{Uconv}}(X) = \exp(iAt) X U, \quad UU^\dagger = I, \tag{5}$$

where $U \in U(d)$ is a unitary operator and $t \in \mathbb{R}$ controls the magnitude of the convolution.

Note that since $A$ is a symmetric matrix, $\exp(iAt)$ is unitary for all values of $t \in \mathbb{R}$ and corresponds to vanilla message passing up to first order: $\exp(iAt) \approx I + iAt + O(t^2)$.

## 3 RELATED WORK

**GNNs for directed graphs.**   Graph Neural Networks for directed graphs have evolved rapidly to capture edge asymmetry beyond simply treating digraphs as undirected. One major line of work extends message-passing neural networks (MPNNs) by separately aggregating incoming and outgoing messages: Dir-GNN performs two distinct neighborhood aggregations, provably matches the expressivity of the Directed Weisfeiler–Lehman test, and yields large gains on heterophilic benchmarks without harming homophilic performance (Rossi et al., 2024). Other models such as DGCN and Di-GCN define directed Laplacians or PPR-based convolutions to incorporate edge

direction (Tong et al., 2020; Zheng et al., 2021). Finally, complex-valued approaches such as MagNet employ a Hermitian adjacency (the Magnetic Laplacian) to encode directionality in the phases of complex embeddings (Zhang et al., 2021).

**Positional encodings, graph transformers, and hybrid models.** By default, transformers treat their inputs as sets, which in the case of graphs means that they can use the node features, but not the topology to make predictions (Kreuzer et al., 2021; Dwivedi & Rampášek, 2021). To remedy this, graph transformers are commonly equipped with positional encodings (PEs), which provide the node features with information about the graph topology. While there is an abundance of positional encodings for undirected graphs (Fesser & Weber, 2023; Rampášek et al., 2022; Dwivedi & Rampášek, 2021; Kreuzer et al., 2021), relatively few encodings have been developed for the directed case. Most prominent among those are PEs based on the transition probabilities of random walks (Geisler et al., 2023) or based on the spectral decomposition of the Magnetic-Laplacian (Geisler et al., 2023; Huang et al., 2024b). PEs and transformers specifically for directed acyclic graphs (DAGs) have also been developed (Luo et al., 2023). Hybrid models are graph transformers that use message-passing GNNs to capture geometric information to complement or even replace positional encodings. Examples of these in the undirected setting include GraphGPS (Dwivedi & Rampášek, 2021), Exphormer (Shirzad et al., 2023), and GRIT (Ma et al., 2023). Notably, these models use PEs *and* MP-GNNs to capture geometric information. Recent work has investigated to what extent MP-GNNs can be used to learn the same geometric information as existing PEs, although these insights are limited to the undirected case (Kanatsoulis et al., 2025).

**Unitary neural networks.** Unitary neural network architectures trace their roots to early efforts at stabilizing deep and recurrent models by enforcing norm-preserving transformations. Initial works applied unitary or orthogonal constraints to RNN weight matrices—using techniques like parameterizations in the Lie algebra or structured matrix factorizations—to mitigate vanishing and exploding gradients and thus capture long-range dependencies in sequence data (Arjovsky et al., 2016; Wisdom et al., 2016; Henaff et al., 2016). Extensions of these ideas to convolutional networks introduced orthogonal and unitary convolutions for image data, often via exponential-map parameterizations of circulant or Lie-group operators, yielding improved training stability in deep CNNs (Sedghi et al., 2018; Li et al., 2019; Trockman & Kolter, 2021; Singla & Feizi, 2021). More recently, graph neural network research has adopted unitary message-passing schemes to prevent representational collapse (oversmoothing) and maintain gradient norms without auxiliary interventions like skip connections or batch normalization (Guo et al., 2022; Qiu et al., 2024; Kiani et al., 2024).

## 4 Dune

The key challenge in adapting unitary convolutions to directed graphs is that their adjacency matrix is in general not symmetric. Consequently, $i\boldsymbol{A}t$ is no longer skew Hermitian for general $t \in \mathbb{R}$, so $\exp(i\boldsymbol{A}t)$ need not be unitary. Our formulation of Dune works around this while also incorporating edge features into the convolution operation. We draw inspiration from (Lezcano-Casado & Martınez-Rubio, 2019) who first proposed a similar Hermitian projection to the one in equation 6, although not in the context of graph machine learning.

### 4.1 Model Definition

**Definition 2.** (Directed unitary GNN with edge features (Dune)). For an arbitrary (not necessarily symmetric) adjacency matrix $\boldsymbol{A} \in \mathbb{R}^{n \times n}$, we consider Hermitian projections of the form

$$\Pi(\boldsymbol{A}) = i\frac{\alpha}{2}\left(\boldsymbol{A} + \boldsymbol{A}^{\dagger}\right) + \frac{\beta}{2}\left(\boldsymbol{A} - \boldsymbol{A}^{\dagger}\right), \tag{6}$$

which is in the Lie algebra of the unitary group for any $\alpha, \beta \in \mathbb{R}$, as a simple calculation shows. In the following we set $\alpha = \beta = \frac{1}{2}$ and perform the unitary convolution as

$$f_{\text{conv}}(\boldsymbol{X}; \boldsymbol{A}) = \exp(\Pi(\boldsymbol{A})t)\boldsymbol{X}\boldsymbol{W}, \tag{7}$$

given a scaling factor $t > 0$ and weights $\boldsymbol{W} \in \mathbb{C}^{d_{in} \times d_{out}}$. exp here denotes the matrix exponential.

Note that a Taylor expansion around $t = 0$ gives

$$\exp\left(\Pi(\boldsymbol{A})t\right) = \boldsymbol{I} + \Pi(\boldsymbol{A})t + O(t^2) = \boldsymbol{I} + \frac{i}{4}(\boldsymbol{A} + \boldsymbol{A}^{\dagger})t + \frac{1}{4}(\boldsymbol{A} - \boldsymbol{A}^{\dagger})t + O(t^2), \tag{8}$$

so a Dune layer sends messages in both directions of a directed edge while still retaining directional information. Assume now that the edges $E$ of the input graph $G = (V, E)$ have edge features $E_{ij} \in \mathbb{R}^{d_E}$. To incorporate these into the convolution operation, we can construct an adjacency matrix: First, we choose a potentially trainable map $f_E : \mathbb{R}^{d_E} \to \mathbb{C}$ and then construct the matrix

$$
[\boldsymbol{A}_E]_{ij} = \begin{cases} f_E(\boldsymbol{E}_{ij}) & \text{if } (i, j) \text{ is an edge} \\ 0 & \text{otherwise} \end{cases} \tag{9}
$$

Then, we perform unitary convolution with the map $\Pi(\boldsymbol{A}_E)$. In a more enhanced version, one can setup a map $f_E : \mathbb{R}^{d_E} \to \mathbb{C}^k$ to have $k$ parallel convolutions if one desires a richer feature space implementation. Unless explicitly stated otherwise, we use a simple linear layer to parametrize $f_E$. Experiments using nonlinear $f_E$ parametrizations in table 8 show no clear advantages. We implement the exponential map in equation 7 using a Taylor approximation. Empirically, truncating after $\sim 10$ terms works well since the approximation error decreases exponentially (Kiani et al., 2024).

### 4.2 DUNE AS AN MPNN

Dune as defined above comes with several theoretical guarantees that make it attractive as a stand-alone message-passing model. We present these results here and defer all proofs to Appendix B. To begin with, a model using Dune layers can be made as expressive as an extension of the 1-WL test to directed graphs (*D-WL*) (Grohe et al., 2021).

**Proposition 3** (Dune Expressivity (informal)). *Consider a directed GNN constructed using the Dune updates in equation 7, where the weight matrices $\boldsymbol{W}^{(k)}$ are injective for all layers $k$ and where any non-linearities $\sigma$ used after equation 7 and the final readout are all injective. Then the model is as expressive as D-WL.*

The proof of this follows from the Taylor expansion in equation 8 and an argument similar to the one presented in (Rossi et al., 2024) for Dir-GNN. From the same paper, it follows that Dune is more powerful than message-passing neural networks that operate on either an undirected version of the graph or that propagate information along only one direction of a directed edge. Unlike Dir-GNN, Dune also comes with guarantees against oversmoothing. Intuitively, oversmoothing describes the phenomenon that the features of neighboring nodes become more similar as the number of network layers increases. This is usually measured via the Dirichlet Energy, which can be defined on directed (and strongly-connected) graphs as follows (Maskey et al., 2023).

**Definition 4.** (Oversmoothing for a $K$-layer GNN on a directed graph). Let $G = (V, E)$ be a directed graph with adjacency matrix $A \in \mathbb{R}^{n \times n}$ and diagonal in-/out-degree matrices $D_{\text{in}}, D_{\text{out}}$. Define the *symmetrically normalized adjacency* $L = D_{\text{in}}^{-\frac{1}{2}} A D_{\text{out}}^{-\frac{1}{2}}$. For node features $\boldsymbol{X} \in \mathbb{R}^{n \times d}$, the (directed) *Dirichlet energy* is

$$
E(\boldsymbol{X}) = \frac{1}{4} \sum_{i,j} a_{ij} \left\| \frac{x_i}{\sqrt{d_i^{\text{in}}}} - \frac{x_j}{\sqrt{d_j^{\text{out}}}} \right\|^2, \qquad d_i^{\text{in}} = (D_{\text{in}})_{ii}, \ d_j^{\text{out}} = (D_{\text{out}})_{jj}. \tag{10}
$$

Consider a standard GNN with $L$ discrete message-passing layers and denote the feature matrix after the $k$-th layer by $\boldsymbol{X}^{(k)}$. Then we may define the *normalized Dirichlet energy* at layer $k$ by

$$
\bar{E}_k := E\left( \frac{\boldsymbol{X}^{(k)}}{\|\boldsymbol{X}^{(k)}\|_F^2} \right). \tag{11}
$$

(Kiani et al., 2024) refer to this as the *Rayleigh quotient*. The GNN is said to *oversmooth* on a graph $G$ and input $\boldsymbol{X}$ if there exist constants $C, C_0 > 0$ such that for every layer $k \geq 1$

$$
\left| \bar{E}_k - \lambda_{\min}(I - L) \right| \leq C_0 e^{-Ck}. \tag{12}
$$

That is, the normalized Dirichlet energy decays exponentially fast toward its minimal attainable value, determined by the spectrum of the directed Laplacian $I - L$.

Like its UniGCN cousin on undirected graphs, Dune can be shown to not suffer from oversmoothing, which opens the way for much deeper models in applications.

**Proposition 5** (Dune never oversmooths)**.** *Assume that the weight matrices $W^{(k)}$ in equation 7 are unitary for all k. Then for any finite directed graph $G = (V, E)$, any step size $t \in \mathbb{R}_{>0}$, any depth $K \in \mathbb{N}$ and any initial node features $X^{(0)}$ that do not already minimize the Rayleigh quotient at initialization, the K-layer Dune model with the update rule in equation 7 does **not** oversmooth, i.e., there do not exist constants $C_0, C > 0$ for which equation 12 holds.*

Dune is therefore an exception among directed graph neural networks. Many other architectures display oversmoothing in experiments, and for some this undesirable behavior can even be proven (DeZoort & Hanin, 2023; Chen et al., 2025). The following proposition illustrates this with the example of DiGCN (Tong et al., 2020). Figure 2 compares this and Dir-GCN (Rossi et al., 2024) experimentally with Dune.

**Proposition 6** (Exponential oversmoothing of DiGCN)**.** *Let $\alpha \in (0, 1)$, let $P = (1-\alpha)\, P_{\mathrm{rw}} + \frac{\alpha}{n}\, \mathbf{1}\mathbf{1}^\top$ be the PageRank transition matrix with stationary distribution $\pi$, and let $\Pi = \mathrm{Diag}(\pi)$, where $\mathbf{1} \in \mathbb{R}^n$ is the all-ones vector. Here $P_{\mathrm{rw}} = D^{-1} A$ is the row-stochastic random-walk matrix built from the adjacency matrix. Define the DiGCN update rule as*

$$S = \tfrac{1}{2}\Big(\Pi^{1/2} P \Pi^{-1/2} + \Pi^{-1/2} P^\top \Pi^{1/2}\Big), \qquad X^{(k+1)} = \sigma\big(S\, X^{(k)} \Theta^{(k)}\big). \qquad (13)$$

*Assume that each $\Theta^{(k)}$ satisfies $\|\Theta^{(k)}\| \leq 1$ and that $\sigma$ is 1-Lipschitz. Then the resulting K-layer DiGCN oversmooths on all Eulerian digraphs (strongly-connected digraphs where all nodes have the same in- and out-degree).*

Finally, and in contrast to conventional GNNs, where the product of contractive Jacobians leads to vanishing gradients (and, less commonly, to exploding ones when spectral norms exceed unity)(Álvaro Arroyo et al., 2025), the Dune update rule 7 guarantees norm preservation at every layer. During backpropagation, the gradient with respect to parameters at layer $i$ involves the product Jacobian

$$J = \prod_{k=i+1}^{K} \frac{\partial X^{(k)}}{\partial X^{(k-1)}}, \qquad (14)$$

whose spectral norm determines whether gradients vanish, explode, or remain stable. Since both the propagator $\exp(t\Pi(A))$ and the weight matrices $W^{(k)}$ are unitary, their operator norms equal one, and thus the layer-wise Jacobians $J_k$ satisfy $\|J_k\|_2 = 1$. Conse-

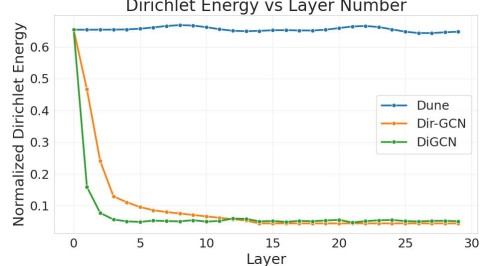

Figure 2: Dirichlet energy comparison on a synthetic Erdos-Renyi graph, generated as in (Geisler et al., 2023).

quently, the gradient norm across $K$ layers is exactly preserved, $\|J\|_2 = 1$, ruling out the exponential decay ($\lambda < 1$) or blow-up ($\lambda > 1$) familiar from standard message-passing networks (at least in the absence of nonlinearities (Arjovsky et al., 2016)). This property shows that vanishing and exploding gradients, typically caused by spectral contraction or expansion in the adjacency and weight matrices, are provably absent in Dune.

### 4.3 Dune in a Hybrid Architecture

We now leave the pure message-passing setting and focus on using Dune in a hybrid model with a graph transformer. In this subsection, we aim to explain which geometric information Dune can capture particularly well for a given input graph. We begin with the following generalization of a directed walk between a pair of nodes.

**Definition 7** (Bidirectional Walk)**.** Let $G = (V, E)$ be a directed graph. A *bidirectional walk* $w = (v_0, v_1, v_2, \ldots)$ is a sequence of nodes where every consecutive pair of nodes forms either a forward edge $(v_i, v_{i+1}) \in E$ or a backward edge $(v_{i+1}, v_i) \in E$. The *length* of a bidirectional walk is the total number of forward and backward edges it contains.

**Definition 8** (Walk Profile)**.** Let $G$ be a directed graph with adjacency matrix $A$. Given two nodes $u, v \in V$, the *walk profile* $\Phi_{u,v}(\ell, k)$ is the number of length-$\ell$ bidirectional walks from $u$ to $v$ that contain exactly $k$ forward edges and $\ell - k$ backward edges.

Using the Magnetic Laplacian, Huang et al. (2024b) introduce a theoretically-grounded positional encoding for directed graphs that provably captures the bidirectional walk profile. To permit the exact recovery of all bidirectional walk counts up to length $\ell$, this Multi-$q$ Magnetic-Laplacian spectral encoding requires $Q \geq \lceil \ell/2 \rceil + 1$ eigendecompositions, which results in a complexity of $O(Q|V|^3)$ (see appendix A.4 for details). We now state informally that Dune captures information about the walk profile in sub-cubic time. Sections B and C in the appendix elaborate further on this.

**Proposition 9** (Dune aggregates information about the walk profiles (informal)). *A depth-K Dune model whose exponential map is implemented via an order T Taylor approximation aggregates weighted sums of bidirectional walk counts up to length $\leq KT$. On directed acyclic graph (DAGs), this recovers the bidirectional walk profiles up to a scalar multiple.*

## 5 EXPERIMENTS

In this section, we experimentally demonstrate the effectiveness of unitary convolutions on directed graphs on a variety of tasks, including node classification and graph regression. In particular, we aim to answer the following questions:

1. Does Dune improve message-passing on directed graphs? And if so, is this mainly due to using unitary convolutions or due to incorporating edge directionality?

2. How does Dune perform as part of a hybrid architecture with graph transformers? Can the information captured by Dune reduce the importance of existing positional encodings, as our theoretical analysis suggests?

In addition to our main experiments, we provide an extensive set of ablations in both the main text and the appendix, which test Dune with hybrid architectures and assess its ability to use edge features.

### 5.1 EXPERIMENTAL DESIGN

To address question 1, we compare Dune against standard MP-GNNs on undirected graphs, as well as state-of-the-art MP-GNNs on directed ones. We conduct an extensive hyperparameter search (see Appendix F for full details) and report the numbers from the original papers whenever they are available and better than the best performance we obtain. For undirected convolutions such as GCN, we convert the input graph into an undirected graph before training. For all models, we record the test set accuracy of the settings with the highest validation accuracy. We accumulate experimental results across 10 random trials and report the mean test accuracy, along with the 95% confidence interval.

For question 2, we follow (Geisler et al., 2023; Huang et al., 2024b) and equip the Structure-Aware Transformer (SAT) (Chen et al., 2022) with different MP-GNN backbones and either no positional encoding ('None') or one of Lap, Maglap computed with only one $q$ value, or the Maglap-Multi-$q$ encoding mentioned in the last section. We report numbers from (Huang et al., 2024b) whenever possible and strictly follow their experimental evaluation protocol and hyperparameter tuning when not. We report the average root mean squared error (RMSE) across 10 random trials along with the 95% confidence interval.

### 5.2 DATASETS

The node-classification experiments draw on seven directed benchmarks, spanning homophilic citation networks (Cora, Citeseer (Bojchevski & Günnemann, 2017), OGBN-Arxiv (Hu et al., 2020)), citation network datasets (Arxiv-Year, Snap-Patents Lim et al. (2021)), and strongly heterophilic graphs (Roman-Empire, Questions (Platonov et al., 2023)). For the Questions dataset we report ROC-AUC, for all other datasets we report accuracy. We evaluate each with its original node features and published train/validation/test splits. Further details can be found in Appendix F.3.

For the hybrid transformer–GNN studies, we use the Open Circuit Benchmark datasets (Dong et al., 2023). These contain 10,000 operational amplifier circuits as directed graphs. The task is to predict the DC gain (Gain), band width (BW) and phase margin (PM) of each circuit. We expect directionality to be useful here because these targets reflect the property of current flows from input nodes to output nodes. The dataset consists of 2-stage amplifiers and 3-stage amplifiers. We follow (Huang et al.,

2024b) and use 2-stage amplifiers, which we randomly split 0.9/0.05/0.05 into train/ val/ test. Also for the hybrid models, we use the HLS dataset (Wu et al., 2022) which contains 18,750 intermediate representation (IR) graphs of C/C++ code after front-end compilation Aho et al. (2006). The dataset provides post-implementation performance metrics on FPGA devices as labels. The task is to predict resource usage: look-up table (LUT) and digital signal processor (DSP) usage. We again follow (Huang et al., 2024b) and randomly select 16570 for training, and 1000 each for validation and testing.

| Model | Cora | Citeseer | OGBN-Arxiv | Arxiv-Year | SNAP-P. | Roman E. | Questions |
|---|---|---|---|---|---|---|---|
| GCN | 84.37 ± 1.52 | 93.37 ± 0.22 | 68.39 ± 0.11 | 46.28 ± 0.39 | 51.02 ± 0.07 | 56.23 ± 0.37 | 76.09 ± 1.27 |
| SAGE | 86.01 ± 1.56 | 94.15 ± 0.61 | 67.78 ± 0.17 | 44.05 ± 0.22 | 52.55 ± 0.10 | 72.05 ± 0.41 | 76.44 ± 0.62 |
| GAT | 86.44 ± 1.45 | 94.53 ± 0.48 | 69.60 ± 0.26 | 45.30 ± 0.23 | 54.28 ± 0.14 | 49.18 ± 1.35 | 77.43 ± 1.20 |
| UniGCN | 86.49 ± 1.56 | 71.13 ± 0.68 | 68.85 ± 0.21 | 55.32 ± 0.22 | 64.67 ± 0.12 | 87.21 ± 0.54 | 79.21 ± 0.81 |
| Di-GCN | 87.35 ± 1.06 | 92.75 ± 0.75 | 65.41 ± 0.12 | 54.13 ± 0.18 | 59.86 ± 0.14 | 54.71 ± 0.32 | 75.82 ± 0.73 |
| MagNet | 85.26 ± 1.05 | 93.38 ± 0.89 | 67.22 ± 0.14 | 60.29 ± 0.27 | 71.33 ± 0.15 | 88.07 ± 0.27 | 77.67 ± 0.78 |
| Dir-GCN | 84.45 ± 1.69 | 93.44 ± 0.59 | 66.66 ± 0.12 | 64.08 ± 0.26 | 73.95 ± 0.06 | 84.54 ± 0.71 | 75.94 ± 1.36 |
| Dir-SAGE | 85.84 ± 2.09 | 94.14 ± 0.65 | 65.14 ± 0.13 | 61.76 ± 0.10 | 70.26 ± 0.14 | 91.23 ± 0.32 | 77.43 ± 0.85 |
| Dir-GAT | 86.21 ± 1.40 | 94.48 ± 0.52 | 66.50 ± 0.16 | 62.47 ± 0.14 | 73.82 ± 0.08 | 82.25 ± 0.14 | 77.19 ± 1.12 |
| NDDGNN | 88.14 ± 1.28 | 94.17 ± 0.58 | 68.76 ± 0.32 | 65.02 ± 0.32 | 74.15 ± 0.04 | 91.76 ± 0.27 | 77.58 ± 1.24 |
| Dune (Ours) | **88.78 ± 1.18** | **94.74 ± 0.81** | **72.16 ± 0.28** | **65.86 ± 0.10** | **75.12 ± 0.14** | **92.58 ± 0.22** | **80.45 ± 0.88** |

Table 2: Undirected (top rows) and directed (bottom rows) message-passing GNNs' performance on directed node classification datasets. Details on baselines can be found in Appendix A.

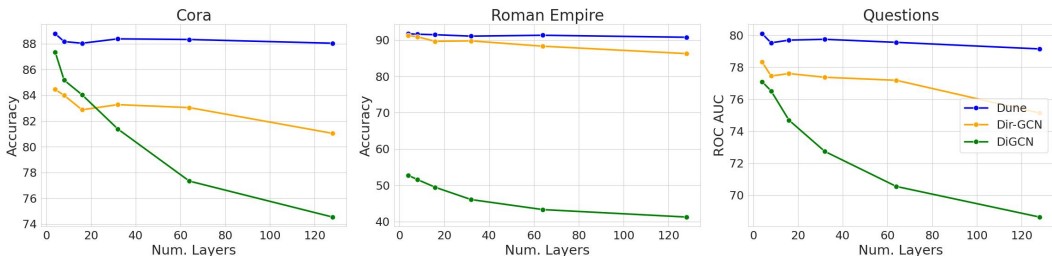

Figure 3: Node classification performance at deeper layers. Higher is better.

## 5.3 RESULTS

We present our experimental results in Tables 2 and 3. Using these, we can answer our original two questions as follows:

1. Dune's superior performance over existing directed GNNs directly reflects the value added by unitary convolutions on top of edge directionality. Across all seven benchmarks, replacing standard message-passing with unitary operations yields consistent gains, with the largest benefits on OGBN-Arxiv (from 68.76 % to 72.16 %) and on Questions (from 77.58 to 80.45). These improvements show that unitary convolutions consistently enhance feature propagation once edge directions are known. Conversely, comparing Dune to UniGCN isolates the benefit of incorporating directionality: performance here improves significantly, i.e. from 64.67 % to 75.12 % on Snap-Patents, and from 87.21 % to 92.58 % on Roman-Empire. We also note that the gains from using Dune are more significant on larger datasets, such as Snap-Patents or OGBN-Arxiv, while on small datasets such as Cora and Citeseer, many models are within reach of each other.

2. When combined with the SAT transformer architecture Chen et al. (2022), Dune turns out to be a much better message-passing backbone than GIN - both directed and undirected - and also outperforms the undirected UniGCN on all five datasets. We also note that with a Dune backbone, the benefits derived from adding positional encodings are much smaller than with other backbones and often become statistically insignificant. This seems to indicate that Dune already captures relevant geometric information, as our theoretical results suggest.

**Additional results** using more hybrid architectures on undirected graphs can be found in Appendix E. Here too, hybrid architectures with unitary message-passing layers significantly outperform commonly-used non-unitary layers, such as GINE or GatedGCN. While our focus in the main text has

been on Dune's performance on directed graphs, we also provide ablations on (undirected) graphs with edge features in Appendix E.4, which establish that Dune effectively captures edge information and outperforms standard UniGCN. Further results on synthetic datasets that investigate to what extent Dune captures geometric information can be found in Appendix E.3.

| Model | PE method | Gain | BW | PM | DSP | LUT |
|-------|-----------|------|------|------|------|------|
| SAT (undirected-GIN) | None | $0.431 \pm 0.023$ | $4.362 \pm 0.134$ | $1.068 \pm 0.036$ | $3.242 \pm 0.205$ | $2.492 \pm 0.172$ |
| | Lap | $0.375 \pm 0.016$ | $4.180 \pm 0.093$ | $1.065 \pm 0.034$ | $3.167 \pm 0.193$ | $2.425 \pm 0.168$ |
| | Maglap-1q (best q) | $0.361 \pm 0.016$ | $4.014 \pm 0.068$ | $1.057 \pm 0.036$ | $3.101 \pm 0.176$ | $2.362 \pm 0.154$ |
| | Maglap-Multi-q | $0.350 \pm 0.004$ | $4.044 \pm 0.153$ | $1.035 \pm 0.025$ | $3.076 \pm 0.240$ | $2.333 \pm 0.147$ |
| SAT (bidirected-GIN) | None | $0.423 \pm 0.018$ | $4.278 \pm 0.102$ | $1.162 \pm 0.038$ | $2.789 \pm 0.142$ | $2.231 \pm 0.117$ |
| | Lap | $0.368 \pm 0.022$ | $4.024 \pm 0.106$ | $1.046 \pm 0.021$ | $2.713 \pm 0.135$ | $2.173 \pm 0.107$ |
| | Maglap-1q (best q) | $0.360 \pm 0.009$ | $3.960 \pm 0.060$ | $1.062 \pm 0.024$ | $2.657 \pm 0.128$ | $2.107 \pm 0.135$ |
| | Maglap-Multi-q | $0.359 \pm 0.008$ | $3.930 \pm 0.069$ | $1.045 \pm 0.012$ | $2.616 \pm 0.151$ | $2.082 \pm 0.099$ |
| SAT (UniGCN) | None | $0.388 \pm 0.019$ | $4.235 \pm 0.098$ | $1.112 \pm 0.030$ | $2.891 \pm 0.149$ | $2.315 \pm 0.126$ |
| | Lap | $0.369 \pm 0.014$ | $4.095 \pm 0.088$ | $1.068 \pm 0.023$ | $2.835 \pm 0.145$ | $2.271 \pm 0.124$ |
| | Maglap-1q (best q) | $0.361 \pm 0.012$ | $4.041 \pm 0.074$ | $1.063 \pm 0.021$ | $2.785 \pm 0.140$ | $2.228 \pm 0.121$ |
| | Maglap-Multi-q | $0.349 \pm 0.011$ | $4.020 \pm 0.072$ | $1.057 \pm 0.019$ | $2.754 \pm 0.132$ | $2.205 \pm 0.114$ |
| SAT (Dune) | None | $0.355 \pm 0.010$ | $3.970 \pm 0.064$ | $1.028 \pm 0.018$ | $2.612 \pm 0.111$ | $2.021 \pm 0.093$ |
| | Lap | $0.352 \pm 0.009$ | $3.945 \pm 0.060$ | $1.025 \pm 0.016$ | $2.570 \pm 0.108$ | $2.015 \pm 0.092$ |
| | Maglap-1q (best q) | $0.350 \pm 0.009$ | $3.932 \pm 0.059$ | $1.023 \pm 0.015$ | $2.558 \pm 0.106$ | $2.007 \pm 0.091$ |
| | Maglap-Multi-q | $0.345 \pm 0.008$ | $3.925 \pm 0.058$ | $1.012 \pm 0.015$ | $2.552 \pm 0.105$ | $1.951 \pm 0.090$ |

Table 3: Graph regression results with hybrid models (SAT + message-passing backbone) and positional encodings. We report mean RMSE (lower is better) and standard deviations over 10 independent runs.

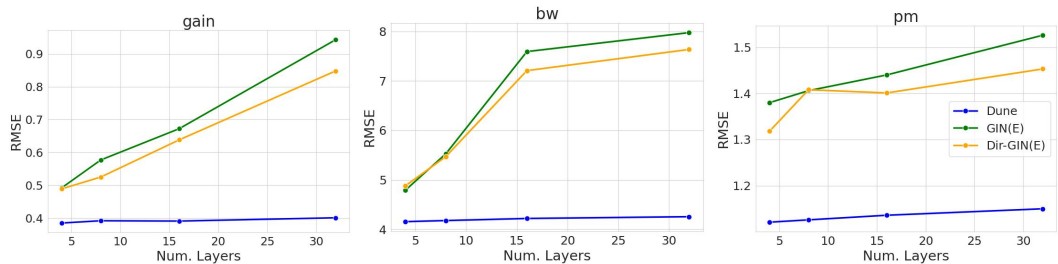

Figure 4: Graph regression performance at deeper layers. Lower is better.

# 6 DISCUSSION

In this paper, we introduced Dune, a unitary graph convolutional network specifically designed for directed graphs, capable of addressing critical problems such as oversmoothing and gradient instabilities that have limited the depth and expressivity of previous directed GNN architectures. Dune leverages norm-preserving, unitary transformations to propagate messages in a manner that prevents representational collapse, ensuring stable training even at considerable depths. Our empirical evaluation demonstrates that Dune consistently outperforms existing models on 12 benchmark datasets, highlighting its effectiveness not only as stand-alone message-passing GNN but also in hybrid architectures involving graph transformers. Importantly, we showed that Dune implicitly encodes geometric information that typically requires PEs.

**Limitations and future work.** Despite these promising results, Dune has several limitations worth highlighting. The computational overhead associated with calculating matrix exponentials for large adjacency matrices, even when projected onto skew-Hermitian forms, can be substantial. A more efficient way to compute (approximately) unitary propagators would be desirable. We also believe that connecting unitary convolutions and positional encodings is a promising area for future investigation. In particular, one might want to use a unitary GNN to explicitly learn PEs in the style of the recently introduced PEARL framework (Kanatsoulis et al., 2025), instead of relying on the inductive biases that come with a unitary GNN in a hybrid architecture. This should allow one to approximately recover the whole walk profile in sub-cubic time, not just part of it. Finally, we believe that directed link prediction, being a significantly more challenging task than node classification or graph regression (He et al., 2025) demands a separate investigation.

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

TABLE OF CONTENTS

# A  FURTHER RELATED LITERATURE

## A.1  GRAPH NEURAL NETWORKS ON UNDIRECTED GRAPHS

**Graph Convolutional Network (GCN) (Kipf & Welling, 2017).**  Let $G = (V, E)$ be an undirected graph with adjacency matrix $A$ and degree matrix $D = \mathrm{diag}(A\mathbf{1})$; its combinatorial Laplacian is $L = D - A$. A GCN stack contains $K$ layers acting on a node-feature matrix $X^{(0)} \in \mathbb{R}^{|V| \times d_0}$. For layer $k \in \{0, \dots, K-1\}$,

$$X^{(k+1)} = \sigma(\hat{A} X^{(k)} W^{(k)}), \qquad \hat{A} = (D + I)^{-\frac{1}{2}}(A + I)(D + I)^{-\frac{1}{2}}, \tag{15}$$

where $W^{(k)} \in \mathbb{R}^{d_k \times d_{k+1}}$ is trainable and $\sigma$ is a point-wise nonlinearity (e.g., ReLU). The composite operator $(\sigma \circ \hat{A})^K$ aggregates information over progressively larger neighborhoods while remaining permutation-equivariant.

**Graph Attention Network (GAT) (Veličković et al., 2017).**  Each GAT layer replaces the fixed propagation matrix $\hat{A}$ with content-dependent attention coefficients. Given node states $x_u^{(k)}$, first compute $z_u = W^{(k)} x_u^{(k)}$. Shared attention parameters $a \in \mathbb{R}^{2d_{k+1}}$ yield unnormalized logits $e_{uv} = \mathrm{LeakyReLU}(a^\top [z_u \| z_v])$ for every edge $(u, v) \in E$ and the self-loop $(v, v)$. Neighborhood-wise soft-max produces $\alpha_{uv} = \exp(e_{uv})/\sum_{w \in \mathcal{N}(v)} \exp(e_{wv})$. The $k$-th layer update is then

$$x_v^{(k+1)} = \sigma\Big( \sum_{u \in \mathcal{N}(v)} \alpha_{uv} z_u \Big), \tag{16}$$

optionally averaged across multiple attention heads, allowing the network to focus dynamically on the most relevant neighbors.

**GraphSAGE (Hamilton et al., 2017).**  GraphSAGE equips each node with an inductive neighborhood aggregator. For layer $k$, sample (or take) the neighbor set $\mathcal{N}(v)$ and compute $\mathrm{AGG}^{(k)}(\mathcal{N}(v))$ using a chosen scheme (mean, max-pool, LSTM, etc.) over $\{x_u^{(k)} : u \in \mathcal{N}(v)\}$. The update rule is

$$x_v^{(k+1)} = \sigma\Big( W^{(k)} \big[ x_v^{(k)} \| \mathrm{AGG}^{(k)}(\mathcal{N}(v)) \big] \Big), \tag{17}$$

followed (optionally) by $\ell_2$ normalization $x_v^{(k+1)} \leftarrow x_v^{(k+1)}/\|x_v^{(k+1)}\|$. After $K$ layers, the model yields scalable, permutation-equivariant representations that capture $K$-hop neighborhood structure without explicitly computing the Laplacian $L$.

## A.2  GRAPH NEURAL NETWORKS ON DIRECTED GRAPHS

**DiGCN (Digraph Inception Convolutional Network) (Tong et al., 2020)**  For a directed graph $G = (V, E)$ with adjacency $A$, set $P = D^{-1}(A + I)$ (row-stochastic with self-loops) and let $\pi^\top$ be its stationary distribution with teleport probability $\alpha$, so that $\Pi = \mathrm{diag}(\pi)$. DiGCN builds an approximate personalized-PageRank magnetic Laplacian

$$L_{\mathrm{appr}} = I - \tfrac{1}{2}(\Pi^{1/2} P \Pi^{-1/2} + \Pi^{-1/2} P^\top \Pi^{1/2}). \tag{18}$$

At layer $k$ ($0 \le k < K$) it forms multi-scale proximity matrices $P^{(0)} = I$, $P^{(1)} = P$, $P^{(r)}$ ($r \ge 2$) by intersecting forward and reverse $r$-hop walks, then computes

$$Z^{(r)} = \begin{cases} X^{(k)} \Theta_0^{(k)}, & r = 0, \\ \tfrac{1}{2}(\Pi^{1/2} P \Pi^{-1/2} + \Pi^{-1/2} P^\top \Pi^{1/2}) X^{(k)} \Theta_1^{(k)}, & r = 1, \\ W_r^{-1/2} P^{(r)} W_r^{-1/2} X^{(k)} \Theta_r^{(k)}, & r \ge 2, \end{cases} \tag{19}$$

with $W_r = \mathrm{diag}(P^{(r)}\mathbf{1})$. An *Inception* fusion operator $\Gamma$ (e.g., sum or concat–linear) combines $\{Z^{(r)}\}_{r=0}^R$, after which $X^{(k+1)} = \sigma(\Gamma(Z^{(0)}, \dots, Z^{(R)}))$. Stacking $K$ layers yields variable-receptive-field, direction-aware convolutions while reserving the symbol $L$ for other Laplacians.

**MagNet (Zhang et al., 2021)** Edge orientation is injected through the *magnetic Laplacian*

$$L_N^{(q)} = I - \big(D_s^{-1/2} A_s D_s^{-1/2}\big) \odot \exp\big(i\Theta^{(q)}\big), \quad \Theta_{uv}^{(q)} = 2\pi q \, (A_{uv} - A_{vu}), \tag{20}$$

where $A_s = \frac{1}{2}(A + A^\top)$ and $D_s$ its degree; $q \in [0, 1/4]$ is learnable. Scaling to $\tilde{L}^{(q)} \in [-1, 1]$, the $k$-th layer applies a Chebyshev filter of order $R$:

$$X^{(k+1)} = \sigma\Big(\sum_{r=0}^{R} T_r\big(\tilde{L}^{(q)}\big) X^{(k)} \Theta_r^{(k)}\Big). \tag{21}$$

Complex ReLU activations maintain phase information, and the real/imaginary parts are concatenated after the final ($K$-th) layer.

**Dir-GCN (Rossi et al., 2024).** Dir-GCN augments message passing with separate inward and outward channels. Define normalized direction matrices $S_\rightarrow = D_{\text{out}}^{-1/2} A \, D_{\text{in}}^{-1/2}$ and $S_\leftarrow = S_\rightarrow^\top$. At layer $k$,

$$\begin{aligned} M_\rightarrow^{(k)} &= S_\rightarrow X^{(k)} W_\rightarrow^{(k)}, \\ M_\leftarrow^{(k)} &= S_\leftarrow X^{(k)} W_\leftarrow^{(k)}, \end{aligned} \tag{22}$$

and the update is

$$X^{(k+1)} = \sigma\big(\alpha M_\rightarrow^{(k)} + (1 - \alpha) M_\leftarrow^{(k)}\big), \qquad \alpha \in [0, 1] \text{ (learnable or fixed)}. \tag{23}$$

Stacking $K$ layers implicitly mixes higher-order operators such as $AA^\top$, enhancing robustness on heterophilic digraphs.

**NDDGNN (Node-Diversity DGNN) (Huang et al., 2024a).** NDDGNN refines Dir-GNN with node-specific gates that consider *neighbor* and *degree* diversity. For each node $v$ at layer $k$ compute Dirichlet energies $e_\rightarrow^{(k)}(v)$ and $e_\leftarrow^{(k)}(v)$, add learned degree embeddings, and apply a softmax to obtain diagonal gates $\Gamma_\rightarrow^{(k)}$ and $\Gamma_\leftarrow^{(k)} = I - \Gamma_\rightarrow^{(k)}$. The update rule is

$$X^{(k+1)} = \sigma\Big(\alpha X^{(k)} + \Gamma_\rightarrow^{(k)} S_\rightarrow X^{(k)} W_\rightarrow^{(k)} + \Gamma_\leftarrow^{(k)} S_\leftarrow X^{(k)} W_\leftarrow^{(k)}\Big), \tag{24}$$

with a consistency regularizer keeping the gates well-behaved. After $K$ layers the model yields direction-adaptive representations without overloading the Laplacian symbol $L$.

A.3 GRAPH TRANSFORMERS

**Structure-Aware Transformer (SAT++) (Chen et al., 2022).** SAT++ augments the vanilla multi-head Transformer encoder with graph-aware inductive biases tailored to code abstract-syntax trees (ASTs). A model comprises $K$ identical layers. At layer $k$ ($1 \le k \le K$) the current node embeddings $H^{(k-1)} \in \mathbb{R}^{n \times d}$ are first processed by a lightweight, three-step, direction-aware GNN that mixes incoming and outgoing messages, yielding query and key bases $\tilde{Q}^{(k)}, \tilde{K}^{(k)}$, while the values remain $V^{(k)} = H^{(k-1)}$. For each attention head $h$ linear projections give $Q_h^{(k)} = \tilde{Q}^{(k)} W_{Q,h}$, $K_h^{(k)} = \tilde{K}^{(k)} W_{K,h}$, and $V_h^{(k)} = V^{(k)} W_{V,h}$. Attention weights are

$$\alpha_h^{(k)} = \text{softmax}\Big(\frac{Q_h^{(k)} K_h^{(k)\top}}{\sqrt{d_h}} + M\Big), \tag{25}$$

where the additive mask $M$ injects edge features and directional positional encodings derived from magnetic-Laplacian and random-walk spectra. Head outputs $O_h^{(k)} = \alpha_h^{(k)} V_h^{(k)}$ are concatenated, projected, and combined with a degree-scaled residual connection followed by LayerNorm to produce $\hat{H}^{(k)}$. A position-wise two-layer feed-forward network with GeLU activation, sparse token dropout, and a second residual-norm pair yields the layer output $H^{(k)}$. After $K$ layers the embedding of the designated *function* node is fed to a linear classifier.

A.4  Positional Encodings on Directed Graphs

**Magnetic-Laplacian spectral encodings (Geisler et al., 2023).**  Let $G = (V, E)$ be a directed graph with adjacency matrix $\boldsymbol{A} \in \{0, 1\}^{|V| \times |V|}$. Define the symmetrized adjacency $\boldsymbol{A}_s = \frac{1}{2}(\boldsymbol{A} + \boldsymbol{A}^\top)$ and its degree matrix $\boldsymbol{D}_s = \mathrm{diag}(\boldsymbol{A}_s \boldsymbol{1})$. For a parameter $q \in [0, 1)$, the *Magnetic Laplacian* is

$$\boldsymbol{L}_U^{(q)} \;=\; \boldsymbol{D}_s \;-\; \boldsymbol{A}_s \odot \exp\!\big(i\Theta^{(q)}\big), \qquad \Theta_{uv}^{(q)} = 2\pi q\, (\boldsymbol{A}_{uv} - \boldsymbol{A}_{vu}). \tag{26}$$

Because $\boldsymbol{L}_U^{(q)}$ is Hermitian, its eigenvectors form an orthonormal basis. Let $\{\phi_\ell\}_{\ell=1}^k$ be the eigenvectors associated with the $k$ smallest eigenvalues. Concatenating their real and imaginary parts yields a $2k$-dimensional node positional encoding $\boldsymbol{z}_v = \big[\mathrm{Re}\,\phi_1(v), \mathrm{Im}\,\phi_1(v), \ldots, \mathrm{Re}\,\phi_k(v), \mathrm{Im}\,\phi_k(v)\big]$ that smoothly rotates along edge directions and preserves orientation information. An optional lightweight two-layer MLP (MagLapNet) may further refine $\boldsymbol{z}_v$ before it is passed to the transformer encoder.

**Multi-$q$ Magnetic-Laplacian spectral encodings (Huang et al., 2024b).**  Building on the Magnetic-Laplacian $\boldsymbol{L}_U^{(q)}$, fix a collection of $Q$ distinct potentials $\{q_j\}_{j=1}^Q \subset [0, 1)$. For each $q_j$, compute the eigenvectors $\{\phi_\ell^{(q_j)}\}_{\ell=1}^k$ of

$$\boldsymbol{L}_U^{(q_j)} \;=\; \boldsymbol{D}_s \;-\; \boldsymbol{A}_s \odot \exp\!\big(i\Theta^{(q_j)}\big), \quad \Theta_{uv}^{(q_j)} = 2\pi\, q_j\, (\boldsymbol{A}_{uv} - \boldsymbol{A}_{vu}),$$

associated with its $k$ smallest eigenvalues. Define the $2k$-dimensional embedding for node $v$ at potential $q_j$ by

$$\boldsymbol{z}_v^{(q_j)} = \big[\mathrm{Re}\,\phi_1^{(q_j)}(v), \mathrm{Im}\,\phi_1^{(q_j)}(v), \ldots, \mathrm{Re}\,\phi_k^{(q_j)}(v), \mathrm{Im}\,\phi_k^{(q_j)}(v)\big].$$

Concatenating these across all $Q$ potentials yields

$$\boldsymbol{z}_v = \big[\boldsymbol{z}_v^{(q_1)}, \ldots, \boldsymbol{z}_v^{(q_Q)}\big] \in \mathbb{R}^{2kQ}.$$

For $Q \geq \lceil L/2 \rceil + 1$, this construction aggregates the phase responses at multiple "frequencies," permitting exact recovery of all bidirectional walk counts up to length $L$ and thereby strictly enhancing expressivity over any single-$q$ scheme.

**Directional random-walk encodings (Geisler et al., 2023).**  Let $\boldsymbol{T} = \boldsymbol{A}\boldsymbol{D}_{\mathrm{out}}^{-1}$ be the forward transition matrix and $\boldsymbol{R} = \boldsymbol{A}^\top \boldsymbol{D}_{\mathrm{in}}^{-1}$ the reverse transition matrix, where $\boldsymbol{D}_{\mathrm{out}}$ and $\boldsymbol{D}_{\mathrm{in}}$ are diagonal out- and in-degree matrices. Self-loops are added to sinks and sources to guarantee ergodicity. For each ordered node pair $(u, v)$ and a fixed horizon $k$, collect the landing probabilities

$$\big[\boldsymbol{R}_{vu}^k, \ldots, \boldsymbol{R}_{vu}, \boldsymbol{T}_{vu}, \ldots, \boldsymbol{T}_{vu}^k\big] \quad \text{and} \quad \mathrm{PPR}_{vu}, \tag{27}$$

where PPR denotes personalized PageRank scores. A two-layer MLP $f_{\mathrm{rw}}^{(2)}$ embeds this pairwise feature vector, and a second MLP $f_{\mathrm{rw}}^{(1)}$ aggregates over all senders $u$ to yield a node-level positional encoding $\zeta(v \,|\, G)$. These encodings capture local directional structure through finite-step walks and global context via PageRank.

# B  Deferred Proofs

**Proposition 10** (Dune Expressivity (formal))**.**  *Let $G = (V, E)$ be a finite directed graph with adjacency matrix $\boldsymbol{A} \in \mathbb{R}^{n \times n}$. Let $\Pi(\boldsymbol{A})$ be the (fixed) skew-Hermitian Dune generator built from $\boldsymbol{A}$. For a node $v$ and a feature matrix $\boldsymbol{X}$, define the* self, out, *and* in *multisets of embeddings as $\mathcal{S}_v^{\mathrm{self}}(\boldsymbol{X}) = \{\!\{\boldsymbol{x}_v\}\!\}$, $\mathcal{S}_v^{\mathrm{out}}(\boldsymbol{X}) = \{\!\{\boldsymbol{x}_u : u \in N_{\mathrm{out}}(v)\}\!\}$, and $\mathcal{S}_v^{\mathrm{in}}(\boldsymbol{X}) = \{\!\{\boldsymbol{x}_u : u \in N_{\mathrm{in}}(v)\}\!\}$, where braces denote* multisets *(order ignored, multiplicities preserved). Fix a finite set of distinct step sizes $\boldsymbol{T} = \{t_1, \ldots, t_m\} \subset \mathbb{R}_{>0}$ and define the stacked multi-t preactivation*

$$\boldsymbol{Z}_{\boldsymbol{T}}(\boldsymbol{X}) \;:=\; \big[\; \boldsymbol{X} \;\big\|\; (\boldsymbol{I} + t_1 \Pi(\boldsymbol{A}))\boldsymbol{X} \;\big\|\; \cdots \;\big\|\; (\boldsymbol{I} + t_m \Pi(\boldsymbol{A}))\boldsymbol{X} \;\big],$$

*so that $[\boldsymbol{Z}_{\boldsymbol{T}}(\boldsymbol{X})]_v$ is the concatenation of $\boldsymbol{x}_v$ and the rows $\big[(\boldsymbol{I} + t_j \Pi(\boldsymbol{A}))\boldsymbol{X}\big]_v$, $j = 1, \ldots, m$. Assume the following* multi-probe injectivity *at each layer: for any feature matrix $\boldsymbol{X}$, the map*

$$v \;\mapsto\; [\boldsymbol{Z}_{\boldsymbol{T}}(\boldsymbol{X})]_v$$

*is an injective function of the pair of multisets $\{\{\boldsymbol{X}_u : u \in N_{\mathrm{out}}(v)\}\}$ and $\{\{\boldsymbol{X}_u : u \in N_{\mathrm{in}}(v)\}\}$ together with $\boldsymbol{X}_v$. Each linear map $\boldsymbol{W}^{(k)} : \mathbb{C}^{(m+1)d_k} \to \mathbb{C}^{d_{k+1}}$ and the nonlinearity $\sigma$ are injective. From the Taylor expansion in equation 8, we use the first-order multi-t Dune update*

$$\boldsymbol{X}^{(k+1)} = \sigma\big(\boldsymbol{Z_T}(\boldsymbol{X}^{(k)})\,\boldsymbol{W}^{(k)}\big), \qquad k = 0, \ldots, K-1.$$

*The directed 1-WL (D-1-WL) update colors nodes via an injective hash of $\big(c^{(k)}(v), \{\{c^{(k)}(u) : u \in N_{\mathrm{out}}(v)\}\}, \{\{c^{(k)}(u) : u \in N_{\mathrm{in}}(v)\}\}\big)$. Initialize $\boldsymbol{X}^{(0)}$ so that $\boldsymbol{x}_v^{(0)} = \boldsymbol{x}_w^{(0)} \iff c^{(0)}(v) = c^{(0)}(w)$. Then for all $k \geq 0$ and all nodes $v, w$,*

$$\boldsymbol{x}_v^{(k)} = \boldsymbol{x}_w^{(k)} \iff c^{(k)}(v) = c^{(k)}(w). \tag{$\star_k$}$$

**Proof of Proposition 10.** We proceed by induction on $k$. Note that the base case ($k = 0$) holds by construction of $\boldsymbol{X}^{(0)}$. For the induction step ($k \to k+1$), assume ($\star_k$) holds. We now prove both directions of ($\star_{k+1}$).

**Forward direction.** Assume $\boldsymbol{x}_v^{(k+1)} = \boldsymbol{x}_w^{(k+1)}$. Injectivity of $\sigma$ and $\boldsymbol{W}^{(k)}$ yields

$$\big[\boldsymbol{Z_T}(\boldsymbol{X}^{(k)})\big]_v = \big[\boldsymbol{Z_T}(\boldsymbol{X}^{(k)})\big]_w.$$

By the multi-probe injectivity assumption, this equality forces the (self, out-, in-) multisets built from $\boldsymbol{x}^{(k)}$ at $v$ and $w$ to be identical. Applying the induction hypothesis ($\star_k$) nodewise converts these multiset equalities of features into multiset equalities of colors, i.e.

$$\{\{c^{(k)}(u) : u \in N_{\mathrm{out}}(v)\}\} = \{\{c^{(k)}(u) : u \in N_{\mathrm{out}}(w)\}\},$$

$$\{\{c^{(k)}(u) : u \in N_{\mathrm{in}}(v)\}\} = \{\{c^{(k)}(u) : u \in N_{\mathrm{in}}(w)\}\},$$

and $c^{(k)}(v) = c^{(k)}(w)$. By the D-1-WL update rule (which hashes these three ingredients injectively), we conclude $c^{(k+1)}(v) = c^{(k+1)}(w)$. This proves the forward implication of ($\star_{k+1}$).

**Reverse direction.** Assume $c^{(k+1)}(v) = c^{(k+1)}(w)$. By the D-1-WL update rule, there is a color-preserving bijection between the multisets

$$\{\{c^{(k)}(u) : u \in N_{\mathrm{out}}(v)\}\} \quad \text{and} \quad \{\{c^{(k)}(u) : u \in N_{\mathrm{out}}(w)\}\},$$

and likewise for in-neighbors, and also $c^{(k)}(v) = c^{(k)}(w)$. By the induction hypothesis ($\star_k$), equality of colors implies equality of features, so the paired bijections lift to

$$\{\{\boldsymbol{x}_u^{(k)} : u \in N_{\mathrm{out}}(v)\}\} = \{\{\boldsymbol{x}_u^{(k)} : u \in N_{\mathrm{out}}(w)\}\},$$

and similarly for in-neighbors, with $\boldsymbol{x}_v^{(k)} = \boldsymbol{x}_w^{(k)}$. By the multi-probe injectivity assumption, these equalities of (in/out/self) multisets imply

$$\big[\boldsymbol{Z_T}(\boldsymbol{X}^{(k)})\big]_v = \big[\boldsymbol{Z_T}(\boldsymbol{X}^{(k)})\big]_w.$$

Applying the injective linear map $\boldsymbol{W}^{(k)}$ and the injective nonlinearity $\sigma$ preserves equality, hence $\boldsymbol{x}_v^{(k+1)} = \boldsymbol{x}_w^{(k+1)}$. This proves the reverse implication of ($\star_{k+1}$).

**Conclusion.** By induction, ($\star_K$) holds. Hence after $K$ layers, Dune produces equal node embeddings if and only if D-1-WL assigns equal colors. With an injective graph-level readout, Dune distinguishes exactly the same pairs of nodes (and graphs) as $K$ rounds of D-1-WL. $\qquad\square$

We note that the above argument closely resembles (Rossi et al., 2024). For more information on the directed Weisfeiler-Leman test and on possible choices for injective activation functions $\sigma$, refer to appendix D in their paper.

**Proof of Proposition 5.** Fix $t > 0$ and set $\boldsymbol{U} := \exp\big(t\,\Pi(\boldsymbol{A})\big)$. A Dune layer updates the features by $\boldsymbol{X}^{(k+1)} = \boldsymbol{U}\,\boldsymbol{X}^{(k)}\,\boldsymbol{W}^{(k)}$, where each trainable weight matrix $\boldsymbol{W}^{(k)} \in \mathrm{U}(d)$ is unitary. Because left- and right-multiplication by unitary matrices preserve the Frobenius norm, $\|\boldsymbol{X}^{(k+1)}\|_F = \|\boldsymbol{X}^{(k)}\|_F$, so $\|\boldsymbol{X}^{(k)}\|_F = \|\boldsymbol{X}^{(0)}\|_F$ for all $k \geq 0$.

For any nonzero feature matrix $X \in \mathbb{C}^{n \times d}$ write the normalized Dirichlet energy as

$$E(X) := \frac{\mathrm{Re}\big[\mathrm{tr}\big(X^\dagger (I - L)\, X\big)\big]}{\|X\|_F^2}, \tag{28}$$

which is real by Proposition 3.6 of Maskey et al. (2023). Proposition 3.3 of the same paper shows that every eigenvalue $\lambda$ of the symmetrically normalized adjacency $L$ satisfies $|\lambda| \le 1$, so each eigenvalue of $I - L$ lies in the closed disk centered at 1 with radius 1. Set

$$\alpha_{\min} := \min_{\lambda \in \lambda(I - L)} \mathrm{Re}\,\lambda, \qquad \alpha_{\max} := \max_{\lambda \in \lambda(I - L)} \mathrm{Re}\,\lambda, \tag{29}$$

noting that $0 \le \alpha_{\min} \le \alpha_{\max} \le 2$. Because a Rayleigh quotient takes values in the numerical range of its matrix,

$$\alpha_{\min} \le E(X) \le \alpha_{\max}, \qquad \text{for every } X \ne 0. \tag{30}$$

Let $E_k := E\big(X^{(k)}\big)$. Unitary similarity yields

$$E_{k+1} = \frac{\mathrm{Re}\big[\mathrm{tr}\big((X^{(k)})^\dagger U^\dagger (I - L)\, U\, X^{(k)}\big)\big]}{\|X^{(k)}\|_F^2}, \tag{31}$$

and since $U^\dagger(I - L)U$ is unitarily similar to $I - L$, the same inequality bounds every $E_k$. Define

$$D_k := E_k - \alpha_{\min}, \qquad M := \alpha_{\max} - \alpha_{\min} \le 2, \tag{32}$$

so $0 \le D_k \le M$ for all $k \ge 0$.

Let $K := \overline{\{U^k : k \ge 0\}} \subset \mathrm{U}(n)$, which is compact. Consider the continuous map

$$f : K \to \mathbb{R}, \qquad f(V) = \frac{\mathrm{Re}\big[\mathrm{tr}\big((X^{(0)})^\dagger V^\dagger (I - L)\, V\, X^{(0)}\big)\big]}{\|X^{(0)}\|_F^2}.$$

Let $M^* := \max_{V \in K} f(V)$ and $m^* := \min_{V \in K} f(V)$ (both attained by compactness). If $X^{(0)}$ is not contained in the $\alpha_{\min}$-eigenspace of $I - L$, then we cannot have $M^* = \alpha_{\min}$. Indeed, if $M^* = \alpha_{\min}$ then $f(V) \equiv \alpha_{\min}$ on $K$, and since $I \in K$ this would give

$$f(I) = \frac{\mathrm{Re}\big[\mathrm{tr}\big((X^{(0)})^\dagger (I - L) X^{(0)}\big)\big]}{\|X^{(0)}\|_F^2} = \alpha_{\min},$$

which forces $(I - L)X^{(0)} = \alpha_{\min} X^{(0)}$, i.e. $X^{(0)}$ lies in the $\alpha_{\min}$-eigenspace—a contradiction. Hence $M^* > \alpha_{\min}$. Pick $V_\star \in K$ with $f(V_\star) \ge \alpha_{\min} + \delta$ for some $\delta > 0$, and extract a subsequence $U^{k_\ell} \to V_\star$. By continuity, $D_{k_\ell} = E_{k_\ell} - \alpha_{\min} \to f(V_\star) - \alpha_{\min} \ge \delta$.

Assume, for contradiction, that a constant $C > 0$ satisfies $D_k \le e^{-Ck}$ for every $k \ge 1$. Then

$$e^{Ck_\ell}\, D_{k_\ell} \ge e^{Ck_\ell} \delta \longrightarrow \infty \qquad (\ell \to \infty), \tag{33}$$

contradicting the bound $D_k \le M^*$. Hence no positive $C$ can satisfy $D_k \le e^{-Ck}$, so a Dune network cannot oversmooth. $\qquad\square$

**Proof of Proposition 6.** The DiGCN update has the form

$$X^{(k+1)} = \sigma\big(S X^{(k)} \Theta^{(k)}\big), \qquad S = \tfrac{1}{2}\Big(\Pi^{1/2} P \Pi^{-1/2} + \Pi^{-1/2} P^\top \Pi^{1/2}\Big),$$

with $\|\Theta^{(k)}\| \le 1$ and $\sigma$ 1-Lipschitz. Since $P = (1 - \alpha)P_{\mathrm{rw}} + \frac{\alpha}{n}\mathbf{1}\mathbf{1}^\top$, its eigenvalues satisfy: one eigenvalue is equal to 1 with eigenvector $\pi$, and all other eigenvalues have modulus at most $1 - \alpha$. By similarity, $\Pi^{1/2} P \Pi^{-1/2}$ has the same spectrum, and hence $S$ has eigenvalues bounded by $1 - \alpha$ on the orthogonal complement of $\pi$. Consequently,

$$\|SZ\|_F \le (1 - \alpha)\|Z\|_F \tag{34}$$

for all feature matrices $Z$ orthogonal to the stationary component. Together with $\|\Theta^{(k)}\| \le 1$ and the 1-Lipschitz property of $\sigma$, each layer is a $\rho$-contraction with constant $\rho = 1 - \alpha < 1$ in Frobenius norm. Iterating this gives

$$\|Z^{(k)}\|_F \le \rho^k \|Z^{(0)}\|_F. \tag{35}$$

Now let $\bar{E}_k = E\left(\frac{X^{(k)}}{\|X^{(k)}\|_F}\right)$ denote the normalized Dirichlet energy of a feature matrix $X^{(k)} = Ua_k + Z^{(k)}$, where $U$ lies in the eigenspace spanned by $\pi$ (assuming non-degeneracy, i.e. that $\lim_{k\to\infty} a_k = a_* \neq 0$). By assumption, the graph $G$ is Eulerian, so the minimizing eigenvectors of $I - S$ and $I - L$ agree. Because $\bar{E}_k$ is a Rayleigh quotient of $I - L$, it is bounded below by $\lambda_{\min}(I - L)$. Using the lower bound $\|X^{(k)}\|_F \geq a_*$, the geometric decay of $Z^{(k)}$ now implies that

$$\left|\bar{E}_k - \lambda_{\min}(I - L)\right| \leq C_0 e^{-Ck} \tag{36}$$

for constants $C = -\log(1 - \alpha)$ and $C_0$, which depends only on the initialization. Hence the normalized Dirichlet energy converges to its minimum at an exponential rate, which is exactly the definition of oversmoothing we use. □

Note that we only used the assumption that $G$ is a Eulerian digraph to align the minimizing subspaces of $I - S$ and $I - L$. The rank collapse phenomenon (see, e.g. (Kiani et al., 2024; DeZoort & Hanin, 2023)), which is closely related to oversmoothing, happens with DiGCN even without this assumption on $G$. We would also like to refer the reader to (Chen et al., 2025), who show oversmoothing for these types of graphs under looser restrictions. Our proposition is only meant to illustrate an example for a specific architecture.

**Proposition 11** (Dune captures information about the walk profiles (formal).). *Let $G$ be a directed graph with adjacency $A$. Fix complex scalars $c_+, c_-$ and set $\Pi(A) = c_+ A + c_- A^\top$. Consider a depth-$K$ Dune stack in which each layer uses the* order-$T$ Taylor truncation of the matrix exponential *with a fixed step $t > 0$. Then the pre-activations at depth $K$ lie in* $\mathrm{span}\{\Pi(A)^\ell X^{(0)} B_\ell : 0 \leq \ell \leq KT\}$ *for some matrices $B_\ell$ depending on the weights and nonlinearity. Consequently, the contribution from a source node $u$ to a target node $v$ at depth $K$ is a weighted sum of bidirectional walk counts* $\{\Phi_{u,v}(\ell, k)\}_{\ell \leq KT}$, *where $\Phi_{u,v}(\ell, k)$ counts length-$\ell$ walks with exactly $k$ forward and $\ell - k$ backward steps:*

$$(\Pi(A)^\ell)_{v,u} = \sum_{k=0}^{\ell} \gamma_{\ell,k} \, \Phi_{u,v}(\ell, k), \qquad 0 \leq \ell \leq KT, \tag{37}$$

*for coefficients $\gamma_{\ell,k}$ determined by $c_\pm$, the Taylor coefficients $t^\ell/\ell!$, and the network parameters.*

**Proof of Proposition 11** One layer applies $P_T(\Pi)$ to the current features, so its pre-activation is a linear combination of $\{\Pi(A)^\ell X^{(r)} : 0 \leq \ell \leq T\}$. Composing $K$ such layers multiplies on the left by degree-$T$ polynomials $K$ times; collecting terms shows every depth-$K$ pre-activation is a linear combination of $\{\Pi(A)^\ell X^{(0)} : \ell \leq KT\}$. Right-side weights and the pointwise nonlinearity only alter the coefficient matrices $B_\ell$; they do not increase the left-multiplicative degree in $\Pi(A)$.

Finally, $\Pi(A) = c_+ A + c_- A^\top$ implies $\Pi(A)^\ell$ expands into words of length $\ell$ in $\{A, A^\top\}$. The $(v, u)$ entry of any such word corresponds to a directed walk from $u$ to $v$ whose steps are "forward" when the factor is $A$ and "backward" when it is $A^\top$; grouping words by the number $k$ of forward steps yields the decomposition above with weights $\gamma_{\ell,k}$. Hence, depth $K$ with order-$T$ propagation aggregates weighted bidirectional walk counts with horizon $\ell \leq KT$. □

A few comments are in order: first, on some directed graphs, this is enough to capture the exact walk profile. For example, on a directed acyclic graph (DAG), Dune's statistic collapses to a single bidirectional count: fix nodes $u \to v$ with shortest directed distance $d(u, v)$. Any length-$d(u, v)$ walk from $u$ to $v$ must be **all forward** ($k = \ell = d(u, v)$); inserting any backward step would force at least two extra steps. Hence for $\ell = d(u, v)$ (and also for $\ell < d(u, v)$, where both sides are zero) the Dune aggregation reduces to a single term $\gamma_{\ell,\ell} \, \Phi_{u,v}(\ell, \ell)$. Hence this recovers the walk profile up to a scalar. This dependency on the topology of the input graph when it comes to capturing information about the bidirectional walk profile also shows empirically (compare DAGs and regular graphs in table 7).

Also, note that a non-unitary directed GNN such as Dir-GCN would require $KT$ layers to recover the same information. Assuming that we want $KT$ to be roughly the diameter of the graph, this quickly becomes problematic on larger graphs due to oversmoothing and exploding or vanishing gradients with non-unitary GNNs. Empirically, we find that this is indeed the case: Dir-GCN performs significantly worse than Dune at predicting shortest and longest directed path distances and fails completely at predicting the walk profile (see table 7).

Finally, we should also point out that the full bidirectional walk profiles can be recovered from the weighted sums of bidirectional walk counts that Dune aggregates using techniques introduced in

(Kanatsoulis et al., 2025). We did not implement this in this paper though as it would go significantly beyond hybrid architectures on directed graphs.

## C  Complexity and Runtime

**Complexity.** Let $n = |V|$ denote the number of nodes. Forming the Hermitian projection $\Pi(A)$ requires computing $A^{\dagger}$ and combining it with $A$, which takes $\Theta(n^2)$ time. The exponential map is then approximated by a Taylor expansion of order $k$,

$$\exp(\Pi(A)) \approx \sum_{j=0}^{k} \tfrac{1}{j!} \Pi(A)^j, \tag{38}$$

and evaluating this expansion explicitly as a matrix requires $k$ successive dense matrix–matrix multiplications of cost $\Theta(n^3)$ each, giving an overall time complexity of $\Theta(kn^3)$. In practice, when applying the exponential to node features $X \in \mathbb{C}^{n \times d}$ instead of computing the full matrix exponential, the cost becomes $k$ multiplications of an $n \times n$ matrix with an $n \times d$ feature matrix, leading to $\Theta(kn^2 d)$ in the dense case or $\Theta(kmd)$ when the adjacency is sparse with $m$ edges.

**Runtime Comparison.** We plot the mean runtime per epoch for a 4-layer model with hidden dimension 64 over 10 independent runs on the seven node classification datasets considered in the main text below. While unitary convolutions in general require more runtime than standard architectures - sometimes by as much as an order of magnitude (OGBN-Arxiv, Arxiv-Year), the overhead that Dune incurs compared to the undirected Uni-GCN is small.

| Model | Cora | Citeseer | OGBN-Arxiv | Arxiv-Year | Snap Patents | Roman Empire | Questions |
|---|---|---|---|---|---|---|---|
| Dir-GCN | 0.005 | 0.004 | 0.015 | 0.017 | 0.191 | 0.018 | 0.006 |
| UniGCN | 0.027 | 0.025 | 0.194 | 0.202 | 0.314 | 0.026 | 0.028 |
| Dune | 0.032 | 0.028 | 0.216 | 0.229 | 0.343 | 0.030 | 0.035 |

Table 4: Runtime comparison across node classification datasets.

## D  Alternative Definitions for Unitary Convolutions on Directed Graphs

In this subsection, we present two alternative ways to define a unitary graph neural network on directed graphs. We did not implement these variants, but believe that they might serve as inspiration for future work.

### D.1  Lie Dune

**Definition (Directed Lie-UniConv).** Let $G = (V, E)$ be a directed graph on $n$ nodes with (possibly asymmetric) adjacency matrix $A \in \mathbb{R}^{n \times n}$. Define the skew-Hermitian projection

$$\Pi(A) \;=\; \frac{i}{4}\,(A + A^{\top}) \;+\; \frac{1}{4}\,(A - A^{\top}) \;\in\; \mathfrak{u}(n),$$

as before and let $W \in \mathbb{C}^{d \times d}$ satisfy $W + W^{\dagger} = 0$. Define the operator

$$g_{\text{conv}}(X) \;=\; \Pi(A)\,X\,W.$$

Then the *directed Lie UniConv* layer is

$$f_{\text{DLie}}(X) = \exp\big(g_{\text{conv}}\big)(X) = \sum_{k=0}^{\infty} \frac{1}{k!}\, g_{\text{conv}}^{(k)}(X),$$

where

$$g_{\text{conv}}^{(0)}(X) = X, \qquad g_{\text{conv}}^{(k)}(X) = \Pi(A)\big(g_{\text{conv}}^{(k-1)}(X)\big)\,W \quad (k \geq 1).$$

We did not implement this, as it exponentiates $W$ by default. If $W$ is skew-hermitian, this results in a unitary weight matrix, which (Kiani et al., 2024) did not always find beneficial, especially on node-level tasks.

## D.2 Dir-GCN Analogue

**Definition (Directed Separable Unitary Graph Convolution).** Let $G = (V, E)$ be a directed graph on $n$ nodes with adjacency matrix $A \in \mathbb{R}^{n \times n}$, which we split into "out-" and "in-" adjacency:

$$A_\to = A, \qquad A_\leftarrow = A^\top.$$

As in Section 4.1, project each into $\mathfrak{u}(n)$ via

$$\Pi(A) = \frac{i}{4}(A + A^\top) + \frac{1}{4}(A - A^\top),$$

so that $\Pi(A_\to) = \Pi(A)$ and $\Pi(A_\leftarrow) = \Pi(A)^\top$ are both skew-Hermitian. Given a diffusion time $t > 0$, a convex-combination weight $\alpha \in [0, 1]$, and learnable (unitary) feature-mixers $U_\to, U_\leftarrow \in U(d)$, the *directed separable UniConv* layer is

$$f_{\mathrm{DirUconv}}(X; A) = \alpha \exp\big(\Pi(A)\,t\big) X U_\to + (1 - \alpha) \exp\big(\Pi(A)^\top t\big) X U_\leftarrow \in \mathbb{C}^{n \times d},$$

which in the small-$t$ limit recovers the usual Dir-GCN-style split aggregation $\alpha A_\to X + (1 - \alpha) A_\leftarrow X$

We did not implement this as it is by construction computationally more costly to compute than our proposed Dune while performing essentially the same convolution operation for small $t$ and appropriate $\alpha$.

# E  Additional Experiments

## E.1 Directed graph regression

| Model | PE method | Gain | BW | PM | DSP | LUT |
|---|---|---|---|---|---|---|
| Undirected-GIN | None | $0.462 \pm 0.023$ | $4.791 \pm 0.122$ | $1.380 \pm 0.030$ | $3.405 \pm 0.204$ | $2.818 \pm 0.190$ |
| | Lap | $0.416 \pm 0.021$ | $4.321 \pm 0.084$ | $1.127 \pm 0.020$ | $2.662 \pm 0.187$ | $1.925 \pm 0.059$ |
| | Maglap-1q (best q) | $0.398 \pm 0.025$ | $4.281 \pm 0.085$ | $1.113 \pm 0.022$ | $2.614 \pm 0.098$ | $2.010 \pm 0.082$ |
| | Maglap-Multi-q | $0.389 \pm 0.017$ | $4.175 \pm 0.115$ | $1.137 \pm 0.004$ | $2.582 \pm 0.133$ | $1.976 \pm 0.089$ |
| Bidirected-GIN | None | $0.466 \pm 0.027$ | $4.879 \pm 0.118$ | $1.318 \pm 0.030$ | $2.637 \pm 0.170$ | $1.814 \pm 0.076$ |
| | Lap | $0.391 \pm 0.007$ | $4.153 \pm 0.160$ | $1.135 \pm 0.035$ | $2.267 \pm 0.126$ | $1.786 \pm 0.072$ |
| | Maglap-1q (best q) | $0.383 \pm 0.002$ | $4.113 \pm 0.052$ | $1.099 \pm 0.020$ | $2.256 \pm 0.144$ | $1.768 \pm 0.090$ |
| | Maglap-Multi-q | $0.371 \pm 0.008$ | $4.051 \pm 0.139$ | $1.116 \pm 0.012$ | $2.207 \pm 0.185$ | $1.735 \pm 0.096$ |
| UniGCN | None | $0.391 \pm 0.018$ | $4.475 \pm 0.096$ | $1.163 \pm 0.027$ | $2.609 \pm 0.113$ | $2.030 \pm 0.102$ |
| | Lap | $0.377 \pm 0.016$ | $4.092 \pm 0.088$ | $1.076 \pm 0.028$ | $2.481 \pm 0.109$ | $1.902 \pm 0.063$ |
| | Maglap-1q (best q) | $0.376 \pm 0.012$ | $4.086 \pm 0.093$ | $1.092 \pm 0.032$ | $2.496 \pm 0.102$ | $1.933 \pm 0.068$ |
| | Maglap-Multi-q | $0.389 \pm 0.022$ | $4.162 \pm 0.125$ | $1.106 \pm 0.024$ | $2.474 \pm 0.094$ | $1.895 \pm 0.074$ |
| Dir-Uni | None | $0.380 \pm 0.011$ | $4.154 \pm 0.083$ | $1.128 \pm 0.024$ | $2.204 \pm 0.091$ | $1.753 \pm 0.081$ |
| | Lap | $0.376 \pm 0.007$ | $4.102 \pm 0.080$ | $1.079 \pm 0.027$ | $2.190 \pm 0.102$ | $1.742 \pm 0.075$ |
| | Maglap-1q (best q) | $0.378 \pm 0.010$ | $4.057 \pm 0.064$ | $1.106 \pm 0.030$ | $2.195 \pm 0.111$ | $1.778 \pm 0.096$ |
| | Maglap-Multi-q | $0.383 \pm 0.008$ | $4.013 \pm 0.078$ | $1.081 \pm 0.019$ | $2.113 \pm 0.094$ | $1.682 \pm 0.063$ |

Table 5: Graph regression results using message-passing architectures. Reported are mean RMSE (lower is better) and standard deviation over 10 independent runs.

| Model | PE method | Gain | BW | PM | DSP | LUT |
|---|---|---|---|---|---|---|
| GPS (UniGCN) | None | $0.398 \pm 0.014$ | $4.428 \pm 0.122$ | $1.201 \pm 0.033$ | $2.781 \pm 0.106$ | $2.153 \pm 0.084$ |
| | Lap | $0.403 \pm 0.028$ | $4.334 \pm 0.108$ | $1.205 \pm 0.030$ | $2.714 \pm 0.095$ | $2.118 \pm 0.064$ |
| | Maglap-1q (best q) | $0.410 \pm 0.017$ | $4.365 \pm 0.113$ | $1.228 \pm 0.036$ | $2.722 \pm 0.113$ | $2.081 \pm 0.105$ |
| | Maglap-Multi-q | $0.407 \pm 0.020$ | $4.352 \pm 0.102$ | $1.194 \pm 0.038$ | $2.689 \pm 0.120$ | $2.052 \pm 0.077$ |
| GPS (Dune) | None | $0.403 \pm 0.012$ | $4.253 \pm 0.128$ | $1.126 \pm 0.028$ | $2.707 \pm 0.088$ | $2.076 \pm 0.072$ |
| | Lap | $0.405 \pm 0.013$ | $4.405 \pm 0.113$ | $1.130 \pm 0.027$ | $2.635 \pm 0.131$ | $2.029 \pm 0.070$ |
| | Maglap-1q (best q) | $0.456 \pm 0.021$ | $4.492 \pm 0.110$ | $1.142 \pm 0.022$ | $2.660 \pm 0.118$ | $2.003 \pm 0.086$ |
| | Maglap-Multi-q | $0.434 \pm 0.018$ | $4.387 \pm 0.116$ | $1.120 \pm 0.020$ | $2.608 \pm 0.102$ | $1.975 \pm 0.062$ |

Table 6: Graph regression results using GraphGPS with a unitary backbone. Reported are mean RMSE (lower is better) and standard deviation over 10 independent runs.

## E.2 TRAINING DYNAMICS AND IMPROVED GENERALIZATION

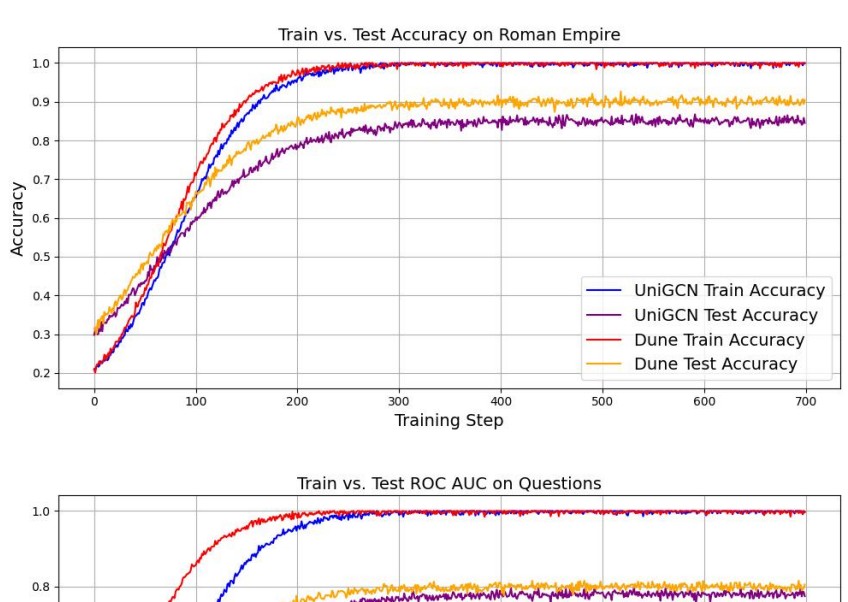

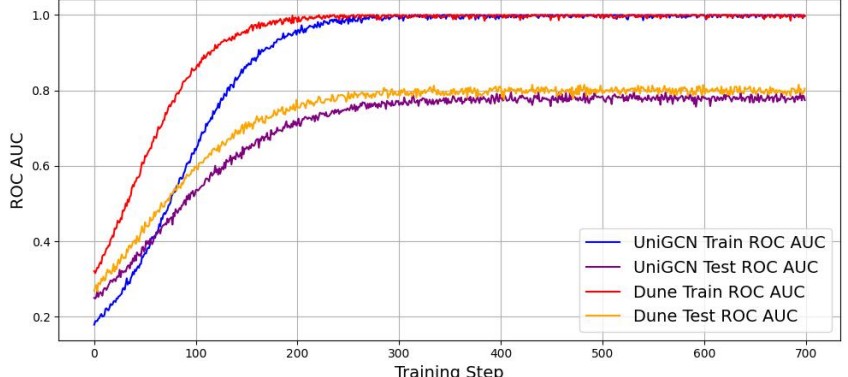

Figure 5: Generalization gaps of UniGCN and Dune on Roman Empire and Questions.

## E.3 POSITIONAL ENCODINGS PLAYGROUND

**Results on synthetic datasets** can be found in Table 7. Here, we follow (Geisler et al., 2023) and (Huang et al., 2024b) and create directed acyclic or regular graphs. We then formulate a regression task by asking a shallow transformer to predict the shortest ('spd') or longest ('lpd') walk distance between two nodes, as well as the complete walk profile ('wp'). The geometric information required for this is provided by either one of four positional encodings or by the forward pass of specifically trained MP-GNN. As Table 7 shows, the model using Dune to derive its geometric information outperforms Lap, SVD, and MagLap-1$q$ positional encodings and the Dir-GCN model while beating MagLap-Multi-q on some datasets.

| PE method | Directed Acyclic Graph | | | Regular Directed Graph | | |
|---|---|---|---|---|---|---|
| | $spd$ | $lpd$ | $wp(4, \cdot)$ | $spd$ | $lpd$ | $wp(4, \cdot)$ |
| Lap | $0.355 \pm 0.001$ | $0.655 \pm 0.002$ | $0.326 \pm 0.001$ | $2.066 \pm 0.005$ | $1.920 \pm 0.000$ | $0.452 \pm 0.001$ |
| SVD | $0.727 \pm 0.001$ | $0.912 \pm 0.001$ | $0.721 \pm 0.000$ | $2.261 \pm 0.002$ | $2.122 \pm 0.007$ | $0.755 \pm 0.000$ |
| MagLap-1q (best q) | $0.124 \pm 0.002$ | $0.432 \pm 0.004$ | $0.040 \pm 0.001$ | $1.533 \pm 0.007$ | $1.493 \pm 0.003$ | $0.132 \pm 0.000$ |
| MagLap-Multi-q | $0.016 \pm 0.000$ | $0.185 \pm 0.036$ | $0.002 \pm 0.000$ | $0.546 \pm 0.068$ | $1.100 \pm 0.007$ | $0.074 \pm 0.001$ |
| Dir-GCN | $0.391 \pm 0.014$ | $0.411 \pm 0.012$ | $3.529 \pm 0.168$ | $1.960 \pm 0.074$ | $3.508 \pm 0.361$ | $5.913 \pm 0.646$ |
| Dune | $0.013 \pm 0.006$ | $0.134 \pm 0.009$ | $0.241 \pm 0.082$ | $0.804 \pm 0.072$ | $1.419 \pm 0.085$ | $0.535 \pm 0.133$ |

Table 7: RMSE results over 3 random seeds for node-pair distance prediction and walk profile reconstruction (lower is better).

| Model | Zinc
Test MAE ↓ | Peptides-func
Test AP ↑ | Peptides-struct
Test MAE ↓ | Coco
Test F1 ↑ | Pascal
Test F1 ↑ |
|---|---|---|---|---|---|
| GCN | 0.367 ± 0.011 | 0.6860 ± 0.0050 | 0.2460 ± 0.0007 | 0.1338 ± 0.0007 | 0.2078 ± 0.0031 |
| GINE | 0.326 ± 0.041 | 0.6621 ± 0.0067 | 0.2473 ± 0.0017 | 0.2125 ± 0.0009 | 0.2718 ± 0.0054 |
| GatedGCN | 0.282 ± 0.015 | 0.6765 ± 0.0047 | 0.2477 ± 0.0009 | 0.2922 ± 0.0018 | 0.3880 ± 0.0040 |
| UniGCN | 0.142 ± 0.022 | 0.7072 ± 0.0035 | 0.2425 ± 0.0009 | 0.2852 ± 0.0016 | 0.3516 ± 0.0070 |
| Dune | 0.093 ± 0.018 | 0.7114 ± 0.0038 | 0.2410 ± 0.0011 | 0.3019 ± 0.0022 | 0.3675 ± 0.0062 |
| Dune(MLP) | 0.091 ± 0.020 | 0.7103 ± 0.0042 | 0.2416 ± 0.0008 | 0.3032 ± 0.0020 | 0.3658 ± 0.0065 |

Table 8: Performance comparison between message-passing architecture on undirected graph datasets (Gómez-Bombarelli et al., 2018; Dwivedi et al., 2022) over 3 random seeds. Dune(MLP) uses a two-layer MLP with gelu activation to process edge features, as opposed to standard Dune's linear layer.

### E.4 MESSAGE-PASSING ON UNDIRECTED GRAPHS

### E.5 HYBRID ARCHITECTURES ON UNDIRECTED GRAPHS

| Model | Zinc
Test MAE ↓ | Peptides-func
Test AP ↑ | Peptides-struct
Test MAE ↓ | Coco
Test F1 ↑ | Pascal
Test F1 ↑ |
|---|---|---|---|---|---|
| GPS(GINE) | 0.083 ± 0.007 | 0.6534 ± 0.0091 | 0.2509 ± 0.0014 | 0.3884 ± 0.0055 | 0.4440 ± 0.0065 |
| GPS(UniGCN) | 0.101 ± 0.014 | 0.6453 ± 0.0086 | 0.2528 ± 0.0017 | 0.3629 ± 0.0067 | 0.4186 ± 0.0092 |
| GPS(Dune) | 0.069 ± 0.010 | 0.6718 ± 0.0097 | 0.2477 ± 0.0015 | 0.3952 ± 0.0061 | 0.4498 ± 0.0071 |

Table 9: Performance comparison between GraphGPS with different backbones on undirected graph datasets over 3 random seeds.

## F EXPERIMENTAL DETAILS

### F.1 HYPERPARAMETER SEARCH PROCEDURE

We carried out a per-dataset grid search, varying the learning rate in {0.001, 0.0005, 0.0001}, hidden dimension in {64, 128, 256, 512}, number of layers in {6, 8, 12, 16, 20, 24}, and dropout rate in {0, 0.2, 0.4, 0.5, 0.7}. Each configuration was trained using the Adam optimizer for up to 10 000 epochs with early stopping on validation accuracy (patience = 500) and no additional regularization.

## F.2 HYPERPARAMETER CONFIGURATIONS

| Model | Cora | Citeseer | OGBN-Arxiv | Arxiv-Year | SNAP-Patents | Roman E. | Questions |
|---|---|---|---|---|---|---|---|
| Learning Rate | 0.001 | 0.001 | 0.0005 | 0.0005 | 0.001 | 0.001 | 0.001 |
| Hidden Dim. | 64 | 512 | 512 | 512 | 64 | 512 | 256 |
| Num. Layers | 8 | 8 | 6 | 6 | 12 | 24 | 16 |
| Dropout | 0.5 | 0.7 | 0 | 0 | 0 | 0.4 | 0.4 |

Table 10: Optimal hyperparameters on node classification datasets with Dune.

| Model | Gain | BW | PM | DSP | LUT |
|---|---|---|---|---|---|
| Learning Rate | 0.001 | 0.001 | 0.001 | 0.004 | 0.004 |
| Hidden Dim. | 54 | 78 | 54 | 78 | 62 |
| Num. Layers | 3 | 3 | 3 | 4 | 3 |
| Dropout | 0.2 | 0.3 | 0.2 | 0.1 | 0.1 |

Table 11: Optimal hyperparameters on graph regression datasets with Dune.

## F.3 DATASETS

| Dataset | # Nodes | # Edges | # Feat. | # C | Unidirectional Edges | Edge hom. |
|---|---|---|---|---|---|---|
| Citeseer-Full | 4,230 | 5,358 | 602 | 6 | 99.61% | 0.949 |
| Cora-ML | 2,995 | 8,416 | 2,879 | 7 | 96.84% | 0.792 |
| OGBN-Arxiv | 169,343 | 1,166,243 | 128 | 40 | 99.27% | 0.655 |
| Arxiv-Year | 169,343 | 1,166,243 | 128 | 40 | 99.27% | 0.221 |
| SNAP-Patents | 2,923,922 | 13,975,791 | 269 | 5 | 99.98% | 0.218 |
| Roman-Empire | 22,662 | 44,363 | 300 | 18 | 65.24% | 0.050 |

Table 12: Statistics of the node classification datasets used in this paper.

## G HARDWARE CONFIGURATION

All experiments in this paper were run on a single Nvidia A100 GPU with 80GB RAM and CUDA version 12.9.

