# OpenReview forum: "Unitary Convolutions for Message-passing and Positional Encodings on Directed Graphs"
_ICLR.cc/2026/Conference — Submitted to ICLR 2026_

### Official Review · Reviewer_GaaG · 2025-10-29

**Soundness:** 3
**Presentation:** 3
**Contribution:** 2
**Rating:** 2
**Confidence:** 4

**Summary:**

This paper investigates how to construct unitary matrices as aggregation operators on directed graphs within the existing UniConv framework. Inspired by the work of Lezcano-Casado et al., the authors adapt their approach to the field of graph learning and propose the Dune model. The method can be theoretically proven to avoid the over-smoothing problem. The authors provide experimental results on both node classification and graph regression tasks, demonstrating the superior performance of Dune compared with the baselines mentioned in the paper.

**Strengths:**

- The paper is overall well-written and easy to follow.
- The authors provide a theoretical analysis to support their claims.
- The model can achieve competitive performance on both homophilic and heterophilic graphs.

**Weaknesses:**

- Complexity
    - Due to the use of the exp(A) operation, the aggregation operator is dense, leading to excessively high model complexity and weak scalability on large-scale datasets.
    - The authors should provide a runtime and memory comparison with simpler models, such as GCN.
- Claim
    - The paper claims to address the problem of stable training; however, it does not provide convincing results to substantiate this claim.
    - The authors claim that their method enables model training with up to 100 layers. However, according to the ablation study, the performance of Dune decreases as the number of layers increases. The current experimental results do not demonstrate any clear advantage of such deep configurations.
- Experiment
    - In the over-smoothing comparisons, such as Dirichlet energy, the authors should at least include the performance of the baseline UniConv to demonstrate that the contribution regarding over-smoothing is novel, rather than an inherent property of the original framework.
    - In the node classification experiments, the authors should consider including comparisons with models such as graph transformers.

- All the figures in the paper are quite blurry. It is recommended that the authors replace them with higher-resolution versions.

**Questions:**

- Given the high complexity of the proposed method, it is unclear how the authors conducted experiments on the SNAP-Patent dataset. Is a mini-batch training strategy employed?

---

> ### Author Response · Authors · 2025-11-19
>
> > Due to the use of the $\exp(A)$ operation, the aggregation operator is dense, leading to excessively high model complexity and weak scalability on large-scale datasets. The authors should provide a runtime and memory comparison with simpler models, such as GCN.
>
> We thank the reviewer for voicing this important concern. We would first like to point out that we only use an order $T$ Taylor approximation for the $\exp(A)$ operation. While we followed [1] in the original manuscript and always used $T=10$, lower values of $T$ are possible and can even result in better performance, especially on homophilous datasets, as the additional results below show:
>
> |         | DUNE (T=2)       | DUNE (T=4)       | DUNE (T=6)       | DUNE (T=8)       | DUNE (T=10)      |
> |---------------|------------------|------------------|------------------|------------------|------------------|
> | Citeseer      | 94.02 ± 0.68     | 95.39 ± 0.62     | 94.61 ± 0.60     | 94.88 ± 0.71     | 94.74 ± 0.81     |
> | OGBN-Arxiv    | 71.69 ± 0.39     | 72.85 ± 0.44     | 72.53 ± 0.32     | 72.04 ± 0.35     | 72.16 ± 0.28     |
> | Questions     | 77.94 ± 0.80     | 78.46 ± 0.83     | 79.27 ± 0.76     | 80.62 ± 0.83     | 80.45 ± 0.88     |
>
> Here, all other hyperparameters are kept as reported in the original manuscript. We now provide the runtime and memory comparisons asked for by the reviewer for various values of $T$:
>
> Peak memory consumption (in MB) with 4 layers and hidden dim. 256. Lower is better.
>
> |         | Dir-GCN  | Dir-GAT  | DUNE (T=2) | DUNE (T=4) | DUNE (T=6) | DUNE (T=8) | DUNE (T=10) |
> |---------------|-----------|-----------|-------------|-------------|-------------|-------------|--------------|
> | Citeseer      | 146.29    | 276.69    | 216.45      | 249.50      | 282.55      | 315.73      | 348.78       |
> | OGBN-Arxiv    | 4063.40   | 17978.97  | 14062.54    | 15390.09    | 16718.63    | 18046.28    | 19374.21     |
> | Questions     | 1241.49   | 3254.66   | 2760.74     | 3144.82     | 3528.46     | 3912.92     | 4296.32      |
>
> Runtime per training epoch in seconds with 4 layers and hidden dim. 256. Lower is better:
>
> |         | Dir-GCN   | Dir-GAT   | DUNE (T=2) | DUNE (T=4) | DUNE (T=6) | DUNE (T=8) | DUNE (T=10) |
> |---------------|------------|------------|-------------|-------------|-------------|-------------|--------------|
> | Citeseer      | 0.00451    | 0.00851    | 0.00593     | 0.00987     | 0.01350     | 0.01880     | 0.02791      |
> | OGBN-Arxiv    | 0.05994    | 0.10627    | 0.14925     | 0.22573     | 0.33003     | 0.54054     | 0.86207      |
> | Questions     | 0.01688    | 0.02514    | 0.01954     | 0.02728     | 0.04143     | 0.05327     | 0.09671      |
>
> The peak memory consumption of Dune is generally comparable to Dir-GAT. For small to moderate values of T, Dune’s runtime is comparable to Dir-GAT, although this is dataset dependent and training Dune can be significantly slower for large values of T. We would like to emphasize though that our implementation is far from optimal, so runtime improvements are certainly possible.
>
> We will expand Appendix C in the current manuscript to include the above results and discussion.
>
> > The paper claims to address the problem of stable training; however, it does not provide convincing results to substantiate this claim. The authors claim that their method enables model training with up to 100 layers. However, according to the ablation study, the performance of Dune decreases as the number of layers increases. The current experimental results do not demonstrate any clear advantage of such deep configurations.
>
> We thank the reviewer for voicing this concern. We politely disagree with the notion that we do not provide substantial evidence to support the claim that Dune remains stable at large depths. Figures 3 and 4 both show that Dune maintains its performance even up to 128 layers. The apparent performance decreases pointed out by the reviewer are well below standard deviation with the exception of the Questions dataset at 128 layers. We will visualize standard deviations in the final manuscript to make this point clearer.
>
> We also thank the reviewer for raising the second point about datasets where deep configurations are advantageous. In response to this, we present additional results on the LRGB datasets below that expand on the LRGB experiments in the original manuscript (Appendix E). In particular, we note that here, performance indeed improves with depth when using DUNE. Non-unitary architectures such as GCN do not benefit from extreme depth, as expected.

---

> > ### Author Response · Authors · 2025-11-19
> >
> > Peptides-func (AP):
> > | Model   | 8 Layers         | 16 Layers        | 32 Layers        | 64 Layers        |
> > |----------|------------------|------------------|------------------|------------------|
> > | GCN      | 0.6864 ± 0.0050  | 0.6907 ± 0.0047  | 0.6429 ± 0.0054  | Diverges         |
> > | UniGCN   | 0.7072 ± 0.0035  | 0.7096 ± 0.0044  | 0.7125 ± 0.0040  | 0.7163 ± 0.0047  |
> > | DUNE     | 0.7108 ± 0.0042  | 0.7114 ± 0.0038  | 0.7183 ± 0.0044  | 0.7221 ± 0.0050  |
> >
> > Pascal (F1):
> > | Model   | 8 Layers         | 16 Layers        | 32 Layers        | 64 Layers        |
> > |----------|------------------|------------------|------------------|------------------|
> > | GCN      | 0.2062 ± 0.0034  | 0.2037 ± 0.0038  | 0.1374 ± 0.0047  | Diverges         |
> > | UniGCN   | 0.3482 ± 0.0058  | 0.3516 ± 0.0070  | 0.3628 ± 0.0067  | 0.3673 ± 0.0074  |
> > | DUNE     | 0.3641 ± 0.0047  | 0.3675 ± 0.0062  | 0.3789 ± 0.0060  | 0.3870 ± 0.0072  |
> >
> > The advantages of incorporating edge features become apparent when comparing DUNE with UniGCN. We will include these results in Appendix E in the final manuscript.
> >
> > > In the over-smoothing comparisons, such as Dirichlet energy, the authors should at least include the performance of the baseline UniConv to demonstrate that the contribution regarding over-smoothing is novel, rather than an inherent property of the original framework.
> >
> > We believe there is a misunderstanding here. As pointed out by the reviewer themselves in their summary of our paper, and as stated in the abstract of the current manuscript, UniConv provably avoids over-smoothing on undirected graphs:
> >
> > “Unitary graph convolutions (UniConv) provably prevent representational collapse and oversmoothing, but cannot incorporate edge directionality or edge features.”
> >
> > Our DUNE model makes it possible to integrate edge directionality and edge features while maintaining unitarity. This makes DUNE the first directed graph neural network to provably mitigate over-smoothing.
> >
> > Plotting the Dirichlet energy with UniConv would require symmetrizing the graph adjacency matrix, precisely because UniConv cannot operate on directed graphs. Only on this now undirected graph would the unitary convolution matrix used by UniConv then ensure that the Dirichlet energy stays approximately constant.
> >
> > DUNE’s inability to avoid this symmetrization step and operate on the original directed graph results in clear performance boosts over UniGCN on all 7 node classification datasets, for example.
> >
> > > In the node classification experiments, the authors should consider including comparisons with models such as graph transformers.
> >
> > We thank the reviewer for this suggestion. We did not originally include graph transformers (GTs) in Table 2 as GTs seem to generally underperform message-passing neural networks on node-level tasks and their costly attention mechanism often limits them to smaller graphs. We nevertheless provide results with two common graph transformers and different backbones below:
> >
> > | Model                 | Cora           | Citeseer       | Roman Empire    | Questions      |
> > |------------------------|----------------|----------------|-----------------|----------------|
> > | GraphGPS (GIN)         | 82.48 ± 0.93   | 92.27 ± 1.07   | 82.52 ± 0.66    | 71.93 ± 1.54   |
> > | Exphormer (GIN)        | 82.69 ± 1.27   | 91.42 ± 1.30   | 83.28 ± 0.58    | 72.74 ± 1.23   |
> > | GraphGPS (bidirected-GIN)     | 83.45 ± 1.19   | 92.12 ± 1.22   | 87.10 ± 0.74    | 74.17 ± 1.46   |
> > | Exphormer (bidirected-GIN)    | 83.68 ± 1.35   | 92.39 ± 1.37   | 87.75 ± 0.80    | 73.84 ± 1.32   |
> > | DUNE                   | 88.78 ± 1.18   | 94.74 ± 0.81   | 92.58 ± 0.22    | 80.45 ± 0.88   |
> >
> > We would be happy to provide further results with other GTs if the reviewer believes these would be informative. Our current plan is to include these additional results in Appendix E in the final manuscript.
> >
> > > All the figures in the paper are quite blurry. It is recommended that the authors replace them with higher-resolution versions.
> >
> > We thank the reviewer for raising this concern. We will ensure that all figures in the final manuscript are high-resolution.
> >
> > > Given the high complexity of the proposed method, it is unclear how the authors conducted experiments on the SNAP-Patent dataset. Is a mini-batch training strategy employed?
> >
> > Yes. The size of the SNAP-Patents dataset makes the use of mini-batching with GAT, Di-GCN, MagNet, Dir-GAT, and with DUNE necessary. We used T=10 for the Taylor approximation in DUNE. We will include this information in the Experimental Details section in the final manuscript.
> >
> > [1] Kiani, B., Fesser, L. and Weber, M., 2024. Unitary convolutions for learning on graphs and groups. Advances in Neural Information Processing Systems, 37, pp.136922-136961.

---

> > > ### Comment · Reviewer_GaaG · 2025-11-26
> > >
> > > I sincerely thank the authors for their detailed responses. Based on the new results and clarifications, several concerns remain:
> > > - Claim to address the problem of stable training: If the authors tend to emphasize the stability of training with different numbers of layers, the claim should be strictly conditioned.
> > > - Comparison on LRGB: More advanced baselines should be included, such as Exphormer and DRew. I am also curious why GCN performs better than the results reported in the original LRGB paper. Were any additional tricks or setting modifications employed?
> > > - Inherent property of UniConv or novel contribution:
> > >    - Even though UniConv cannot be directly applied to directed graphs, the authors may consider converting directed graphs into undirected ones for Dirichlet energy comparison. Otherwise, the main contribution may be viewed as extending UniConv to directed graphs, rather than further addressing over-smoothing.
> > >    - The same is true for the stable training. On peptides-func and pascal-voc, UniConv already demonstrates the ability to achieve better performance with increased depth. Therefore, the ability to train deep models stably appears to be inherited from UniConv, rather than a novel contribution introduced by Dune.

---

> > > > ### Author Response · Authors · 2025-11-27
> > > >
> > > > > Claim to address the problem of stable training: If the authors tend to emphasize the stability of training with different numbers of layers, the claim should be strictly conditioned.
> > > >
> > > > We thank the reviewer for highlighting this - we agree the claim should be stated more precisely. In the final manuscript we will separate (i) the theoretical result (bounded gradients / no oversmoothing under our unitary propagation) from (ii) the empirical observation that the model trains reliably at high depth on the benchmarks we report, and we will explicitly bound the empirical depth range (e.g., “stable up to 128 layers in our experiments”) rather than implying generality beyond what we measured. We’ll reflect this conditioning in the abstract, contributions, and results sections to avoid overstatement.
> > > >
> > > > Could we ask the reviewer to confirm that this is indeed what they meant by “strictly conditioned” and “stability of training with different numbers of layers”?
> > > >
> > > > > Comparison on LRGB: More advanced baselines should be included, such as Exphormer and DRew. I am also curious why GCN performs better than the results reported in the original LRGB paper. Were any additional tricks or setting modifications employed?
> > > >
> > > > We thank the reviewer for pointing this out. We report results for Exphormer and DRew below (taken from [1], whose evaluation protocol and hyperparameters we follow for all LRGB results).
> > > >
> > > > | Model     | Peptides-Func        | Pascal              |
> > > > |-----------|------------------------|----------------------|
> > > > | GCN       | 0.6907 ± 0.0047        | 0.2062 ± 0.0034      |
> > > > | DRew      | 0.7150 ± 0.0044        | 0.3314 ± 0.0024      |
> > > > | Exphormer | 0.6527 ± 0.0043        | 0.3960 ± 0.0027      |
> > > > | UniGCN    | 0.7163 ± 0.0047        | 0.3673 ± 0.0074      |
> > > > | DUNE      | 0.7221 ± 0.0050        | 0.3870 ± 0.0072      |
> > > >
> > > > Dune outperforms DRew on both Peptides-func and Pascal, on the latter dataset by a wide margin. It also significantly outperforms Exphormer on Peptides-func, and achieves only slightly lower performance than Exphormer on Pascal. We will run additional hyperparameter sweeps using both models for the final manuscript to confirm these trends.
> > > >
> > > > Please also note that we follow [1] and report GCN performance on LRGB with residual connections, dropout, and normalization, which significantly boosts performance and makes for a stronger baseline to compare against. Using the original LRGB settings (without these components), we indeed recover the lower GCN numbers mentioned by the reviewer. We will document these details in the appendix of the final manuscript for full transparency.
> > > >
> > > > > Even though UniConv cannot be directly applied to directed graphs, the authors may consider converting directed graphs into undirected ones for Dirichlet energy comparison. Otherwise, the main contribution may be viewed as extending UniConv to directed graphs, rather than further addressing over-smoothing.
> > > >
> > > > We thank the reviewer for clarifying this concern. We will add a control plot on the symmetrized graphs that reports Dirichlet energy vs. depth for GCN and UniGCN, so readers can relate our results to the classical (undirected) notion of oversmoothing. We stress, however, that comparing Dirichlet energies across directed and undirected settings is not apples-to-apples, as the related notions of oversmoothing differ significantly [2].
> > > >
> > > > Our contribution targets the directed case: Dune preserves the Dirichlet energy while incorporating edge features and edge directionality, thereby preventing oversmoothing in that setting. We also want to stress that separate from oversmoothing, using directionality is important for performance (see our empirical results below and in the original manuscript), so converting directed graphs into undirected ones may not be the right thing to do.
> > > >
> > > > > The same is true for the stable training. On peptides-func and pascal-voc, UniConv already demonstrates the ability to achieve better performance with increased depth. Therefore, the ability to train deep models stably appears to be inherited from UniConv, rather than a novel contribution introduced by Dune.
> > > >
> > > > We thank the reviewer for pointing this out. Since the LRGB datasets consist of undirected graphs, we indeed expect UniConv to also do well here. The improved performance that Dune displays here is most likely due to more effective edge-feature modeling. We originally chose LRGB to address the reviewer’s concern because the benchmark was explicitly designed to require the effective modeling of long-range interactions, which necessitates either global attention or a large number of layers.
> > > >
> > > > As an example for a dataset where Dune’s performance increases with depth, while UniGCN plateaus or decays, we provide additional ablations on the DSP and LUT datasets below. These datasets consist of directed graphs with edge features. We report RMSE, so lower is better, and use the same evaluation protocol and hyperparameters as in the original manuscript.

---

> > > > > ### Author Response · Authors · 2025-11-27
> > > > >
> > > > > DSP:
> > > > >
> > > > > | Model          | 4 Layers            | 8 Layers            | 16 Layers           | 32 Layers           |
> > > > > |----------------|----------------------|----------------------|----------------------|----------------------|
> > > > > | Bidirected-GIN | 2.637 ± 0.170        | 2.348 ± 0.143        | 2.466 ± 0.158        | 2.695 ± 0.135        |
> > > > > | UniGCN         | 2.609 ± 0.113        | 2.552 ± 0.109        | 2.583 ± 0.127        | 2.634 ± 0.117        |
> > > > > | DUNE           | 2.204 ± 0.091        | 2.145 ± 0.096        | 2.112 ± 0.088        | 2.086 ± 0.084        |
> > > > >
> > > > > LUT:
> > > > >
> > > > > | Model          | 4 Layers            | 8 Layers            | 16 Layers           | 32 Layers           |
> > > > > |----------------|----------------------|----------------------|----------------------|----------------------|
> > > > > | Bidirected-GIN | 1.814 ± 0.076        | 1.765 ± 0.084        | 1.869 ± 0.091        | 1.927 ± 0.080        |
> > > > > | UniGCN         | 2.030 ± 0.102        | 1.974 ± 0.095        | 2.050 ± 0.078        | 2.061 ± 0.089        |
> > > > > | DUNE           | 1.753 ± 0.081        | 1.712 ± 0.072        | 1.684 ± 0.075        | 1.645 ± 0.068        |
> > > > >
> > > > > We believe that this further illustrates the benefits of using Dune in the directed graphs setting, where UniGCN falls short, and would be happy to include these results in the appendix of the final manuscript.
> > > > >
> > > > > [1] Kiani, B., Fesser, L. and Weber, M., 2024. Unitary convolutions for learning on graphs and groups. Advances in Neural Information Processing Systems, 37, pp.136922-136961.
> > > > >
> > > > > [2] Sohir Maskey, Raffaele Paolino, Aras Bacho, and Gitta Kutyniok. A fractional graph laplacian approach to oversmoothing. Advances in Neural Information Processing Systems, 36:13022–13063, 2023.

---

### Official Review · Reviewer_kVV9 · 2025-10-31

**Soundness:** 3
**Presentation:** 3
**Contribution:** 3
**Rating:** 6
**Confidence:** 3

**Summary:**

The paper introduces Dune, a directed unitary GNN with edge features that extends UniConv, a NeurIPS 2024 paper entitled "Unitary convolutions for learning on graphs and groups". Unlike UniConv, Dune incorporates edge directionality and edge features while still preventing representational collapse and oversmoothing. Its unitary operator keeps gradient norms bounded across layers, enabling very deep architectures. Dune can also be combined with graph transformers, where its wave-like propagation provides positional information without relying on random-walk or Laplacian encodings. The authors prove Dune avoids exponential oversmoothing in directed GNNs and demonstrate state-of-the-art results on 12 benchmarks, with performance gains of up to 18 percentage points and stable training beyond 100 layers. This positions unitary convolutions as a scalable, geometry-aware approach for deep learning on directed graphs.

**Strengths:**

- Combines edge direction and edge features
- Study shows that both unitary convolutions and edge directionality help
- Detailed analysis, ablation studies

**Weaknesses:**

- Oversmoothing is addressed due to UniConv, so it does not come from the newly introduced approach
- Edge feature modeling remains shallow
- The computational overhead is significant and thus cannnot easily scale to large graphs. The authors mention that in the limitations Section. Nevertheless, it is a very important weakness and an approximated solution would definitely help the approach.

**Questions:**

- Is Dune able to capture richer edge semantics (e.g. multi-relational edges, higher-order edge dependencies)?
- Why does Dune work better for some datasets and not for all?

Suggestions
- Add a reference of Table 1 within the text, and state what is exactly shown there (e.g. expressivity)

---

> ### Author Response · Authors · 2025-11-18
>
> > Oversmoothing is addressed due to UniConv, so it does not come from the newly introduced approach
>
> The reviewer is correct in so far as mitigating feature collapse/ over-smoothing is an inherent ability of unitary neural networks and not limited to our DUNE model. For example, as stated in the abstract of the current manuscript, UniConv provably avoids over-smoothing **on undirected graphs**:
>
> “Unitary graph convolutions (UniConv) provably prevent representational collapse and oversmoothing, but cannot incorporate edge directionality or edge features.”
>
> Our DUNE model makes it possible to integrate edge directionality and edge features while maintaining unitarity. This makes DUNE the **first directed graph neural network to provably mitigate over-smoothing**.
>
> DUNE can avoid the symmetrization step required for UniGCN and operate on the original directed graph instead. This results in clear performance boosts over UniGCN on all 7 node classification datasets, for example.
>
> > Edge feature modeling remains shallow
>
> We thank the reviewer for voicing this concern. We would like to point out the ablations in Appendix E.4 of our original submission which indicate that using a parametrization for the edge feature map $f_E$ that is more complicated than a linear map does not yield clear advantages.
>
> Nevertheless, since reviewer 1L3e voiced similar concerns, we provide some additional results on edge modeling below:
>
> Graph regression results on the HLS datasets. Lower is better. “Linear” and “MLP” refers to the way the edge feature map in equation 9 was parametrized.
>
> | Model                   | DSP            | LUT            |
> |--------------------------|----------------|----------------|
> | UniGCN                   | 2.609 ± 0.113  | 2.030 ± 0.102  |
> | DUNE, no edges           | 2.316 ± 0.104  | 1.884 ± 0.095  |
> | DUNE, linear edges       | 2.204 ± 0.091  | 1.753 ± 0.081  |
> | DUNE, MLP edges          | 2.219 ± 0.088  | 1.738 ± 0.086  |
> |                          |                |                |
> | SAT (UniGCN)             | 2.891 ± 0.149  | 2.315 ± 0.126  |
> | SAT (DUNE, no edges)     | 2.728 ± 0.137  | 2.179 ± 0.113  |
> | SAT (DUNE, linear edges) | 2.612 ± 0.111  | 2.021 ± 0.093  |
> | SAT (DUNE, MLP edges)    | 2.590 ± 0.120  | 2.046 ± 0.098  |
>
> We again find that incorporating edge features is crucial but doing so with more than a linear map does not result in consistent performance gains. (All other hyperparameters from the original paper were kept)
>
> > The computational overhead is significant and thus cannot easily scale to large graphs. The authors mention that in the limitations Section. Nevertheless, it is a very important weakness and an approximated solution would definitely help the approach.
>
> We respectfully disagree with the notion that DUNE cannot scale to larger graphs. Please note that we evaluate Dune on three node classification datasets with more than one million edges (OGBN-Arxiv, Arxiv-Year, and SNAP-Patents) to demonstrate the scalability of our method. We are not aware of any larger directed commonly studied graph datasets. The computational cost can therefore be substantial, as pointed out in the limitations section, but is not prohibitive.
>
> Furthermore, as explained in Appendix C, the computational cost of Dune can be reduced by using a smaller value of $T$ in the Taylor approximation of the exponential map. In our original submission, we used $T=10$ as suggested by [1] and did not ablate on smaller values of $T$. It turns out that this is not optimal. We provide some ablations on the importance of $T$ below, with all other hyperparameters kept as reported in the original manuscript:
>
> |         | DUNE (T=2)       | DUNE (T=4)       | DUNE (T=6)       | DUNE (T=8)       | DUNE (T=10)      |
> |---------------|------------------|------------------|------------------|------------------|------------------|
> | Citeseer      | 94.02 ± 0.68     | 95.39 ± 0.62     | 94.61 ± 0.60     | 94.88 ± 0.71     | 94.74 ± 0.81     |
> | OGBN-Arxiv    | 71.69 ± 0.39     | 72.85 ± 0.44     | 72.53 ± 0.32     | 72.04 ± 0.35     | 72.16 ± 0.28     |
> | Questions     | 77.94 ± 0.80     | 78.46 ± 0.83     | 79.27 ± 0.76     | 80.62 ± 0.83     | 80.45 ± 0.88     |
>
> On homophilous datasets, such as Citeseer and OGBN-Arxiv, performance actually increases as we reduce $T$ (up to a point), while on the heterophilous Questions dataset, a larger value of $T$ is beneficial. Varying $T$ therefore provides practitioners with the ability to adjust DUNE to the requirements of their datasets and to also reduce computational burden: the overall complexity of the model scales linearly in $T$, so reducing it, especially on homophilous datasets can reduce runtime significantly.

---

> > ### Author Response · Authors · 2025-11-18
> >
> > > Is Dune able to capture richer edge semantics (e.g. multi-relational edges, higher-order edge dependencies)?
> >
> > Yes. Note, for example, that the Open Circuit and HLS benchmarks used in Table 3 already probe rich edge semantics. In particular, every edge carries both an edge type and additional, edge-level features that are usually scalar-valued. We would also like to point out that these tasks, by definition, require DUNE to model the higher-order dependencies between multiple directed edges, such as motifs. Please also see our earlier ablations on the importance of edge features in the DSP and LUT datasets that are part of the HLS benchmark.
> >
> > In the final manuscript, we will provide additional clarifications on the nature of the datasets used and the importance of edge directionality and edge features. In particular, we will use section 5.2 to clarify the crucial role of edge features for the tasks in Table 3.
> >
> > > Why does Dune work better for some datasets and not for all?
> >
> > We interpret “work better” as performance improvements over non-unitary GNNs. We believe that the variation we observe in Table 2 can be at least partially explained by homophily and heterophily in the underlying datasets. On homophilous node classification tasks, oversmoothing is generally less of a problem than on heterophilous ones. This relates back to our earlier discussion of the optimal value of $T$.
> >
> > > Add a reference of Table 1 within the text, and state what is exactly shown there (e.g. expressivity)
> >
> > We thank the reviewer for suggesting this. We will make this adjustment in the final manuscript.
> >
> > [1] Kiani, B., Fesser, L. and Weber, M., 2024. Unitary convolutions for learning on graphs and groups. Advances in Neural Information Processing Systems, 37, pp.136922-136961.

---

> ### Comment · Reviewer_kVV9 · 2025-11-24
>
> I thank the authors for their comments and increase my score accordingly.

---

### Official Review · Reviewer_1L3e · 2025-11-01

**Soundness:** 3
**Presentation:** 3
**Contribution:** 2
**Rating:** 4
**Confidence:** 2

**Summary:**

This paper proposes a method that solves the key problems of oversmoothing and gradient when applying standard GNNs to directed graphs. It keeps gradient norms bounded at any number of layers, provably avoids oversmoothing and gradient pathologies, allowing it to be trained at depths beyond 100 layers.

**Strengths:**

1. The paper introduces a novel and elegant method—a Hermitian projection—to generalize unitary convolutions to asymmetric (directed) graphs
2. It provably guarantees the proposed method avoids the exponential oversmoothing and vanishing/exploding gradients.
3. This paper discusses the limitations of its computation cost.

**Weaknesses:**

1. The mechanism for incorporating edge features is not as theoretically integrated as the core topological framework
2. The matrix exponential is approximated with $T \approx 10$. How sensitive is the model's performance to the order $T$ of the Taylor approximation? Could $T$ be significantly reduced (e.g., $T=2$ or $3$) to achieve a runtime closer to standard GNNs while still retaining the majority of the benefits over non-unitary models?
3. Lack of experiments on both graphs with strong directionality and rich, meaningful edge features

**Questions:**

How sensitive is the model's performance to the order $T$ of the Taylor approximation?

---

> ### Author Response · Authors · 2025-11-18
>
> > The mechanism for incorporating edge features is not as theoretically integrated as the core topological framework.
>
> We think this may be a misunderstanding as our approach is rather flexible while maintaining theoretical guarantees. In DUNE, edge attributes $E_{ij}$ are mapped by a learnable feature map $f_E$ to form an edge-weighted adjacency $A_E$​, after which we apply the same skew-Hermitian projection and unitary propagator. Theoretical guarantees (unitarity, norm/gradient preservation, mitigation of oversmoothing) hold for any parameterization of $f_E$. For example, $f_E$​ could be an MLP. Empirically, we find that a simple linear $f_E$​ performs on par with a two-layer MLP and opted to keep it simple. Could the reviewer clarify what they mean by theoretically integrated perhaps?
>
> > The matrix exponential is approximated with T. How sensitive is the model's performance to the order  of the Taylor approximation? Could T be significantly reduced (e.g., $T=2$ or $T=3$) to achieve a runtime closer to standard GNNs while still retaining the majority of the benefits over non-unitary models?
>
> We would like to thank the reviewer for bringing up this excellent point. We had originally focused on using $T=10$ or greater, as suggested by [1]. It turns out that this is not optimal. We provide some ablations on the importance of $T$ below, with all other hyperparameters kept as reported in the original manuscript:
>
> |         | DUNE (T=2)       | DUNE (T=4)       | DUNE (T=6)       | DUNE (T=8)       | DUNE (T=10)      |
> |---------------|------------------|------------------|------------------|------------------|------------------|
> | Citeseer      | 94.02 ± 0.68     | 95.39 ± 0.62     | 94.61 ± 0.60     | 94.88 ± 0.71     | 94.74 ± 0.81     |
> | OGBN-Arxiv    | 71.69 ± 0.39     | 72.85 ± 0.44     | 72.53 ± 0.32     | 72.04 ± 0.35     | 72.16 ± 0.28     |
> | Questions     | 77.94 ± 0.80     | 78.46 ± 0.83     | 79.27 ± 0.76     | 80.62 ± 0.83     | 80.45 ± 0.88     |
>
> We find the results remarkable in that on homophilous datasets, such as Citeseer and OGBN-Arxiv, performance actually increases slightly as we reduce $T$ (up to a point), which we believe is due to a certain amount of oversmoothing not being harmful on these datasets. Hence enforcing strict unitarity by using a large value of $T$ might not be necessary. Note that this approximate unitarity is still better than non-unitary GNNs.
>
> By contrast, performance drops sharply on the heterophilous Questions datasets once we reduce $T$ to $6$ or lower. Intuitively, this again makes sense as oversmoothing on heterophilous datasets is clearly harmful. Varying $T$ therefore provides practitioners with the ability to adjust DUNE to the requirements of their datasets and to also reduce computational burden: the overall complexity of the model scales linearly in $T$, so reducing it, especially on homophilous datasets, can reduce runtime significantly.
>
> We are very grateful to the reviewer for raising this point and will include the above ablation and further results related to the role of $T$ (e.g. runtime) in the final manuscript.
>
> > Lack of experiments on both graphs with strong directionality and rich, meaningful edge features
>
> We believe that there is a misunderstanding here. We provide experimental results on 12 directed graph datasets and on 10 datasets with edge features (with 5 of these 10 in Table 3). The datasets used for the experiments in Table 3 in particular involve both directed edges and meaningful edge features without which performance on these tasks drops significantly. We provide some additional ablations to reinforce this point below:
>
> Graph regression results on the HLS datasets. Lower is better. “Linear” and “MLP” refers to the way the edge feature map in equation 9 was parametrized.
>
> | Model                   | DSP            | LUT            |
> |--------------------------|----------------|----------------|
> | UniGCN                   | 2.609 ± 0.113  | 2.030 ± 0.102  |
> | DUNE, no edges           | 2.316 ± 0.104  | 1.884 ± 0.095  |
> | DUNE, linear edges       | 2.204 ± 0.091  | 1.753 ± 0.081  |
> | DUNE, MLP edges          | 2.219 ± 0.088  | 1.738 ± 0.086  |
> |                          |                |                |
> | SAT (UniGCN)             | 2.891 ± 0.149  | 2.315 ± 0.126  |
> | SAT (DUNE, no edges)     | 2.728 ± 0.137  | 2.179 ± 0.113  |
> | SAT (DUNE, linear edges) | 2.612 ± 0.111  | 2.021 ± 0.093  |
> | SAT (DUNE, MLP edges)    | 2.590 ± 0.120  | 2.046 ± 0.098  |
>
> In the final manuscript, we will provide additional clarifications on the nature of the datasets used and the importance of edge directionality and edge features. In particular, we will use section 5.2 to clarify the crucial role of edge features for the tasks in Table 3.
>
> > How sensitive is the model's performance to the order T of the Taylor approximation?
>
> Please see our comment above.

---

> > ### Author Response · Authors · 2025-11-18
> >
> > [1] Kiani, B., Fesser, L. and Weber, M., 2024. Unitary convolutions for learning on graphs and groups. Advances in Neural Information Processing Systems, 37, pp.136922-136961.

---

> > > ### Author Response · Authors · 2025-11-27
> > >
> > > Dear reviewer 1L3e, we appreciate your time and careful review of our paper.
> > >
> > > Since we have now provided individual, point-by-point responses - including ablations on the role of the order $T$ of the Taylor approximation - could you please confirm that these address your earlier concerns?
> > >
> > > If anything remains unclear or unaddressed, we would be grateful if you could flag it so we can clarify promptly during the remainder of the discussion period. Thank you again for helping us improve the paper.

---

### Official Review · Reviewer_a2Gp · 2025-11-07

**Soundness:** 3
**Presentation:** 2
**Contribution:** 3
**Rating:** 6
**Confidence:** 4

**Summary:**

This paper presents Dune, a directed unitary GNN that incorporates edge directionality and edge features while preserving the stability guarantees of unitary graph convolutions. By leveraging unitary transformations, Dune maintains bounded gradient norms at arbitrary depths, enabling deep training without oversmoothing. It can also serve as a component in hybrid architectures (e.g., graph transformers), where its wavelike propagation implicitly encodes positional information, reducing the need for external positional encodings. Theoretical analysis proves Dune’s resistance to exponential oversmoothing and gradient explosion, while extensive experiments on 12 directed-graph benchmarks demonstrate state-of-the-art performance and scalability beyond 100 layers.

**Strengths:**

S1: The asymmetric adjacency matrix of a directed graph is projected into a Hermitian form, making the exponential operator exp unitary. Following the UniConv paradigm, this leads to a new directed graph convolution framework.

S2: A key property of unitary transformations is that they inherently prevent oversmoothing, gradient explosion, and vanishing during message passing. This has been theoretically proven, enabling the effective training of directed GNNs at very deep layer depths.

S3: The experiment examines whether performance gains in directed message passing arise from the convolution mechanism itself or from modeling edge directionality. It also investigates whether the proposed Dune can achieve an effect comparable to positional encoding in capturing geometric structural information.

**Weaknesses:**

W1: Although unitary transformations are theoretically associated with stability, the paper provides little quantitative analysis of computational efficiency (e.g., FLOPs, parameter count) compared with baseline GNNs such as GCN or GraphSAGE, or with other stability-oriented methods like residual connections and normalization. This gap limits the assessment of the model’s practical scalability.

W2: While the paper presents many theoretical analyses, it would benefit from more intuitive explanations or illustrative examples to help readers better understand the proposed method.

W3: The claimed contribution on positional encoding seems only marginally different from existing approaches, and the paper would benefit from a more precise articulation of its advantages.

**Questions:**

Q1: The theoretical proof of oversmoothing avoidance primarily considers static, homophilic graphs. How does this guarantee extend to heterophilic or dynamic graphs? Could the authors provide supplementary analyses or theoretical bounds for these cases?

Q2: Can the proposed Dune model effectively alleviate the oversquashing problem in graph learning?

---

> ### Author Response · Authors · 2025-11-18
>
> > Although unitary transformations are theoretically associated with stability, the paper provides little quantitative analysis of computational efficiency (e.g., FLOPs, parameter count) compared with baseline GNNs such as GCN or GraphSAGE, or with other stability-oriented methods like residual connections and normalization. This gap limits the assessment of the model’s practical scalability.
>
> We thank the reviewer for voicing their concern. Please note that Appendix C provides results on complexity and wall-clock time. Please also note that we evaluate Dune on three node classification datasets with more than one million edges (OGBN-Arxiv, Arxiv-Year, and SNAP-Patents) to demonstrate the scalability of our method. We are not aware of any larger commonly studied directed graph datasets.
>
> For the comparison between unitary convolutions and residual connections or layer normalization, we now provide the following ablations, which we will include in the appendix of the final manuscript.
>
> **Roman Empire (Accuracy):**
> | Model   | Skip + Norm.       | Skip only         | Norm. only        | Neither          |
> |----------|--------------------|-------------------|-------------------|------------------|
> | GCN      | 56.23 ± 0.37       | 55.96 ± 0.46      | 55.02 ± 0.59      | 54.68 ± 0.66     |
> | Dir-GCN  | 84.54 ± 0.71       | 83.61 ± 0.68      | 83.17 ± 0.82      | 83.03 ± 0.95     |
> | DUNE     | 92.58 ± 0.22       | 92.45 ± 0.26      | 92.36 ± 0.39      | 92.12 ± 0.44     |
>
>
> **Questions (ROC-AUC):**
> | Model   | Skip + Norm.       | Skip only         | Norm. only        | Neither          |
> |----------|--------------------|-------------------|-------------------|------------------|
> | GCN      | 76.09 ± 1.27       | 74.58 ± 1.34      | 73.87 ± 1.38      | 73.19 ± 1.46     |
> | Dir-GCN  | 75.94 ± 1.36       | 75.03 ± 1.39      | 74.82 ± 1.42      | 74.21 ± 1.48     |
> | DUNE     | 80.45 ± 0.88       | 80.10 ± 0.93      | 79.92 ± 0.98      | 79.65 ± 1.04     |
>
> Here, we hold the number of parameters fixed between models. As expected, GCN and Dir-GCN benefit from both skip connections and normalization, while the effect for Dune is generally not statistically significant. This aligns with our prior as unitary neural networks were introduced specifically to guarantee stable gradients and no oversmoothing without skip connections or layer normalization (see *Related Work* section).
>
> > While the paper presents many theoretical analyses, it would benefit from more intuitive explanations or illustrative examples to help readers better understand the proposed method.
>
> We would like to thank the reviewer for pointing this out. We were hoping that Figure 1 might help carry the main ideas of the paper (projection onto the unitary group, capturing of topological features such as walk distances). Would it help if we redid Figure 1, perhaps with a clearer focus on the DUNE architecture? Any suggestions would be most welcome.
>
> > The claimed contribution on positional encoding seems only marginally different from existing approaches, and the paper would benefit from a more precise articulation of its advantages.
>
> We thank the reviewer for the thoughtful comment. We agree the paper should state this more clearly and will clarify this in the final manuscript. Our aim is not to introduce yet another positional encoding, but to show that Dune’s unitary propagation already supplies the orientation-aware, multi-hop signals that directed graph PEs target. Theoretically, the propagator can aggregate weighted bidirectional-walk statistics akin to magnetic-Laplacian PEs, but without repeated eigendecompositions. Empirically, Dune is strong without any PE, and adding Lap/MagLap yields only modest gains, indicating the backbone captures the needed geometry. The advantage, therefore, is reduced reliance on precomputed PEs and a more scalable, end-to-end approach. To the best of our knowledge, this has not been studied in the context of directed graphs before.
>
> > The theoretical proof of oversmoothing avoidance primarily considers static, homophilic graphs. How does this guarantee extend to heterophilic or dynamic graphs? Could the authors provide supplementary analyses or theoretical bounds for these cases?
>
> Our theoretical guarantees extend directly to heterophilic graphs. Proposition 5, for example, makes no assumptions about homo- or heterophily. Please also note that 4 of the 7 node classification datasets we consider (Arxiv-Years, Snap Patents, Roman Empire, and Questions) are heterophilic. Dune outperforms all baselines on these.
>
> Our oversmoothing results similarly extend to graph neural networks on dynamic graphs. For example, one could incorporate Dune into the Roland framework [1] to work with dynamic graphs. While we did not investigate this experimentally, we will include unitary neural networks on dynamic graphs in our future works section.
>
> We would like to thank the reviewer for bringing this up.

---

> > ### Author Response · Authors · 2025-11-18
> >
> > > Can the proposed Dune model effectively alleviate the oversquashing problem in graph learning?
> >
> > Yes. Empirical evidence for this can be found in Appendix E, where we provide additional experiments on the long-range graph benchmark (LRGB) datasets, which are frequently used to study oversquashing. Dune performs significantly better than non-unitary message-passing architectures.
> >
> > A theoretical argument for this should also be possible, for example by using the notion of oversquashing considered in [2]. Oversquashing tends to occur in the presence of a “bottleneck” in the graph tied to the spectral gap of the adjacency matrix $A$, i.e. the difference between the first two eigenvalues. Since Dune applies linear unitary transformations, all eigenvalues of the linear transformation are projected onto the unit circle. Thus, Dune has no notion of a spectral gap, and oversquashing is alleviated. We note, however, that [2] limit themselves to undirected graphs and connections with other definitions of oversquashing might be less direct.
> >
> > [1] You, J., Du, T. and Leskovec, J., 2022, August. ROLAND: graph learning framework for dynamic graphs. In Proceedings of the 28th ACM SIGKDD conference on knowledge discovery and data mining (pp. 2358-2366).
> >
> > [2] Karhadkar, K., Banerjee, P. and Montufar, G., 2023, February. FoSR: First-order spectral rewiring for addressing oversquashing in GNNs. In International Conference on Learning Representations.

---

> > > ### Author Response · Authors · 2025-11-27
> > >
> > > Dear reviewer a2Gp, we appreciate your time and careful review of our paper.
> > >
> > > Since we have now provided individual, point-by-point responses - including ablations on the importance of residual connections and layer normalization - could you please confirm that these address your earlier concerns?
> > >
> > > If anything remains unclear or unaddressed, we would be grateful if you could flag it so we can clarify promptly during the remainder of the discussion period. Thank you again for helping us improve the paper.

---

### Official Review · Reviewer_78Pn · 2025-11-08

**Soundness:** 3
**Presentation:** 1
**Contribution:** 2
**Rating:** 4
**Confidence:** 4

**Summary:**

The paper introduces a novel GNN architecture, namely Dune, designed for directed graphs with edge features through the use of a unitary convolution which provably prevents oversmoothing and exploding/vanishing gradients, allowing for stable training of deep GNNs. Additionally, the authors propose using Dune in hybrid architectures, reducing excessive reliance of graph transformers to positional encodings.

**Strengths:**

1. The model provably avoids exponential oversmoothing and vanishing/exploding gradient problems
2. The paper extends the concept of unitary convolutions to the more complex case of directed graphs

**Weaknesses:**

1. The paper is difficult to read. Several key concepts are introduced at a very high level:
    - Section 4.3 is particularly unclear. It states that Dune can be used in a hybrid model but it is not explained how. I think the node embeddings from the previous layer are first passed through a Dune message-passing step and then the model performs attention over those (while the PE is only added at the first layer), but this is not explained.
    - Overall, the paper is missing intuitive explanations of concepts. For instance, an intuitive explanation of the method, even on a simple 3 node graphs could significantly improve readability.
2. The model is motivated by its stability at extreme depths (100+ layers in Fig. 3). However, the optimal hyperparameter settings for the main benchmark results (Table 10) use only 6-24 layers. This either means that the chosen datasets do not require such depths, and therefore do not showcase the capabilities of the model, or that the model cannot really work with significantly large depths. Assuming it is the former, you should test on datasets where a very large number of layers is necessary for obtaining good results.
3. The empirical evaluation is missing a simple baseline, that is, a standard GCN/SAGE that only aggregates incoming edges (e.g., using $D_{in}^{-1}A$). The paper's "undirected" baselines are weaker as they are forced to use an undirected graph.

**Additionally, note that there is some font issue as the section titles appear to have a different font than the standard iclr template.**

**Questions:**

1.  In the introduction's "common strategies" why not mention the simple approach of aggregating only incoming edges?  This also impact the results because in the experiments you can use a simple gcn baseline that only aggregates incoming edges, without converting the graph into an undirected one.
2. Are you assuming multiple convolutions are stacked one after the other with non-linearities in between? Cause equation 1 cannot represent an entire gnn, but it should be seen as a single gnn layer, with multiple layers interleaved by non linearities, This should be made clear.
3. How does your work compare to Eliasof et al., AAAI 2024. Feature Transportation Improves Graph Neural Networks, which shows that including a direction behavior mitigates oversmoothing?
4. The expressivity proof (Proposition 3/10) requires injective non-linearities and an injective final readout. Were these conditions met in the experiments?
5. What is the wall-clock time, memory overhead and complexity analysis of your method, especially when compared to the standard sparse-matrix multiplication used in baseline gnn methods and Dir-GCN?

---

> ### Author Response · Authors · 2025-11-18
>
> > The paper is difficult to read. Several key concepts are introduced at a very high level: 1) Section 4.3 is particularly unclear. It states that Dune can be used in a hybrid model but it is not explained how. I think the node embeddings from the previous layer are first passed through a Dune message-passing step and then the model performs attention over those (while the PE is only added at the first layer), but this is not explained. 2) Overall, the paper is missing intuitive explanations of concepts. For instance, an intuitive explanation of the method, even on a simple 3 node graphs could significantly improve readability.
>
> 1) The hybrid models we consider such as SAT or GraphGPS are indeed built in the way described by the reviewer. We agree with the reviewer that the section in its current form needs some clarification on this, which we will include in the revised manuscript.
>
> 2) We would like to thank the reviewer for pointing this out. We were hoping that Figure 1 might help carry the main ideas of the paper (projection onto the unitary group, capturing of topological features such as walk distances). Would it help if we redid Figure 1, perhaps with a clearer focus on the DUNE architecture? Any suggestions would be most welcome.
>
> > The model is motivated by its stability at extreme depths (100+ layers in Fig. 3). However, the optimal hyperparameter settings for the main benchmark results (Table 10) use only 6-24 layers. This either means that the chosen datasets do not require such depths, and therefore do not showcase the capabilities of the model, or that the model cannot really work with significantly large depths. Assuming it is the former, you should test on datasets where a very large number of layers is necessary for obtaining good results.
>
> We thank the reviewer for pointing this out. We present additional results on the long-range graph benchmark (LRGB) datasets below that expand on the LRGB experiments in the original manuscript. The LRGB datasets were introduced to test a GNN’s ability to model interactions between distant nodes, which in principle should require either global attention or a large number of message-passing steps (especially on Pascal) [1].  Indeed, we note that here, performance indeed improves with depth when using DUNE. Non-unitary architectures such as GCN do not benefit from extreme depth, as expected.
>
> Peptides-func (AP):
> | Model   | 8 Layers         | 16 Layers        | 32 Layers        | 64 Layers        |
> |----------|------------------|------------------|------------------|------------------|
> | GCN      | 0.6864 ± 0.0050  | 0.6907 ± 0.0047  | 0.6429 ± 0.0054  | Diverges         |
> | UniGCN   | 0.7072 ± 0.0035  | 0.7096 ± 0.0044  | 0.7125 ± 0.0040  | 0.7163 ± 0.0047  |
> | DUNE     | 0.7108 ± 0.0042  | 0.7114 ± 0.0038  | 0.7183 ± 0.0044  | 0.7221 ± 0.0050  |
>
> Pascal (F1):
> | Model   | 8 Layers         | 16 Layers        | 32 Layers        | 64 Layers        |
> |----------|------------------|------------------|------------------|------------------|
> | GCN      | 0.2062 ± 0.0034  | 0.2037 ± 0.0038  | 0.1374 ± 0.0047  | Diverges         |
> | UniGCN   | 0.3482 ± 0.0058  | 0.3516 ± 0.0070  | 0.3628 ± 0.0067  | 0.3673 ± 0.0074  |
> | DUNE     | 0.3641 ± 0.0047  | 0.3675 ± 0.0062  | 0.3789 ± 0.0060  | 0.3870 ± 0.0072  |

---

> > ### Author Response · Authors · 2025-11-18
> >
> > > The empirical evaluation is missing a simple baseline, that is, a standard GCN/SAGE that only aggregates incoming edges. The paper's "undirected" baselines are weaker as they are forced to use an undirected graph.
> >
> > We thank the reviewer for suggesting this and provide the requested additional results below. For the sake of completeness, we also include a baseline that only aggregates ‘outgoing’ edges (GCN-Out and Sage-Out). In general, we find that on some datasets this idea indeed outperforms our previous undirected baselines and even baselines that aggregate information in both directions. On some datasets, it also leads to dramatic performance drops, however (OGBN-Arxiv, Roman Empire).
> >
> > | Model     | Cora           | Citeseer       | OGBN-Arxiv     | Arxiv-Year     | Roman Empire    | Questions      |
> > |------------|----------------|----------------|----------------|----------------|-----------------|----------------|
> > | GCN        | 84.37 ± 1.52   | 93.37 ± 0.22   | 68.39 ± 0.11   | 46.28 ± 0.39   | 56.23 ± 0.37    | 76.09 ± 1.27   |
> > | SAGE       | 86.01 ± 1.56   | 94.15 ± 0.61   | 67.78 ± 0.17   | 44.05 ± 0.22   | 72.05 ± 0.41    | 76.44 ± 0.62   |
> > | GCN-In     | 84.53 ± 1.65   | 93.86 ± 0.24   | 63.84 ± 0.25   | 52.56 ± 0.51   | 52.26 ± 0.42    | 76.30 ± 1.53   |
> > | Sage-In    | 85.26 ± 1.49   | 93.92 ± 0.67   | 65.77 ± 0.22   | 53.38 ± 0.29   | 71.28 ± 0.46    | 77.13 ± 0.71   |
> > | GCN-Out    | 85.07 ± 1.76   | 93.33 ± 0.35   | 24.04 ± 0.67   | 51.90 ± 0.44   | 44.14 ± 0.67    | 74.96 ± 1.41   |
> > | Sage-Out   | 85.47 ± 1.44   | 92.80 ± 0.58   | 52.01 ± 0.39   | 49.76 ± 0.38   | 79.76 ± 0.43    | 75.67 ± 0.85   |
> > | Dir-GCN    | 84.45 ± 1.69   | 93.44 ± 0.59   | 66.66 ± 0.12   | 64.08 ± 0.26   | 84.54 ± 0.71    | 75.94 ± 1.36   |
> > | Dir-Sage   | 85.84 ± 2.09   | 94.14 ± 0.65   | 65.14 ± 0.13   | 61.76 ± 0.10   | 91.23 ± 0.32    | 77.43 ± 0.85   |
> > | DUNE       | 88.78 ± 1.18   | 94.74 ± 0.81   | 72.16 ± 0.28   | 65.86 ± 0.10   | 92.58 ± 0.22    | 80.45 ± 0.88   |
> >
> > > Additionally, note that there is some font issue as the section titles appear to have a different font than the standard iclr template.
> >
> > We thank the reviewer for pointing this out and will make sure that the section titles in the final manuscript agree with ICLR guidelines.
> >
> > > In the introduction's "common strategies" why not mention the simple approach of aggregating only incoming edges? This also impacts the results because in the experiments you can use a simple gcn baseline that only aggregates incoming edges, without converting the graph into an undirected one.
> >
> > Please see our comment above. We will include this in both the introduction and in the experiments section (Table 2).
> >
> > > Are you assuming multiple convolutions are stacked one after the other with non-linearities in between? Cause equation 1 cannot represent an entire gnn, but it should be seen as a single gnn layer, with multiple layers interleaved by non linearities, This should be made clear.
> >
> > We thank the reviewer for pointing this out, Equation 1 is indeed only describing a single convolutional layer, not an entire GNN. We will include a sentence about this (and non-linearities) in the revised manuscript.
> >
> > > How does your work compare to Eliasof et al., AAAI 2024. Feature Transportation Improves Graph Neural Networks, which shows that including a direction behavior mitigates oversmoothing?
> >
> > In short, Eliasof et al. show that adding directional transport via a learnable advection term (with operator-splitting diffusion/reaction) improves propagation on directed/heterophilic graphs and empirically mitigates oversmoothing. DUNE follows a similar intuition (edge direction matters), but attains directionality through a unitary propagator built from a skew-Hermitian generator of the adjacency, which guarantees no oversmoothing and no vanishing/exploding gradients.
> >
> > We thank the reviewer for bringing this up. We will be sure to cite the paper and include it as related literature.
> >
> > > The expressivity proof (Proposition 3/10) requires injective non-linearities and an injective final readout. Were these conditions met in the experiments?
> >
> > Yes, in part. We compared leaky ReLU (which is injective) with GeLU and ReLU for the graph regression tasks in Table 3, but found all three to perform within standard deviation of each other.
> >
> > Hence all experimental results are reported with ReLU, which ensures comparability with other works such as Rossi et al. [2] by limiting the architectural modifications to the convolution. To the best of our knowledge, the existing literature (especially on undirected graphs) uses injective non-linearities mainly as a tool to establish expressivity via the WL hierarchy, not for experiments on real-world data. We likewise include Proposition 3 primarily to allow for a rigorous comparison with [2].

---

> ### Author Response · Authors · 2025-11-19
>
> > What is the wall-clock time, memory overhead and complexity analysis of your method, especially when compared to the standard sparse-matrix multiplication used in baseline gnn methods and Dir-GCN?
>
> Our original manuscript contained a short discussion of complexity and wall-clock time in Appendix C. We expand upon this below and in particular ablate on the order of the Taylor approximation $T$ (original results were reported with $T=10$). We find that wall-clock time and memory overhead, which are usually significant in unitary neural networks [3] decrease visibly as we decrease $T$, while affecting performance only marginally and sometimes even improving it. We believe that this is due to varying levels of homophily and heterophily in the datasets considered, as over-smoothing tends to be less harmful on homophilous graphs.
>
> DUNE accuracy (ROC-AUC) with different values of $T$. All other hyperparameters kept fixed and the values reported in the original manuscript:
>
> |         | DUNE (T=2)       | DUNE (T=4)       | DUNE (T=6)       | DUNE (T=8)       | DUNE (T=10)      |
> |---------------|------------------|------------------|------------------|------------------|------------------|
> | Citeseer      | 94.02 ± 0.68     | 95.39 ± 0.62     | 94.61 ± 0.60     | 94.88 ± 0.71     | 94.74 ± 0.81     |
> | OGBN-Arxiv    | 71.69 ± 0.39     | 72.85 ± 0.44     | 72.53 ± 0.32     | 72.04 ± 0.35     | 72.16 ± 0.28     |
> | Questions     | 77.94 ± 0.80     | 78.46 ± 0.83     | 79.27 ± 0.76     | 80.62 ± 0.83     | 80.45 ± 0.88     |
>
> Peak memory consumption (in MB) with 4 layers and hidden dim. 256. Lower is better:
>
> |         | Dir-GCN  | Dir-GAT  | DUNE (T=2) | DUNE (T=4) | DUNE (T=6) | DUNE (T=8) | DUNE (T=10) |
> |---------------|-----------|-----------|-------------|-------------|-------------|-------------|--------------|
> | Citeseer      | 146.29    | 276.69    | 216.45      | 249.50      | 282.55      | 315.73      | 348.78       |
> | OGBN-Arxiv    | 4063.40   | 17978.97  | 14062.54    | 15390.09    | 16718.63    | 18046.28    | 19374.21     |
> | Questions     | 1241.49   | 3254.66   | 2760.74     | 3144.82     | 3528.46     | 3912.92     | 4296.32      |
>
> Runtime per training epoch in seconds with 4 layers and hidden dim. 256. Lower is better:
>
> |         | Dir-GCN   | Dir-GAT   | DUNE (T=2) | DUNE (T=4) | DUNE (T=6) | DUNE (T=8) | DUNE (T=10) |
> |---------------|------------|------------|-------------|-------------|-------------|-------------|--------------|
> | Citeseer      | 0.00451    | 0.00851    | 0.00593     | 0.00987     | 0.01350     | 0.01880     | 0.02791      |
> | OGBN-Arxiv    | 0.05994    | 0.10627    | 0.14925     | 0.22573     | 0.33003     | 0.54054     | 0.86207      |
> | Questions     | 0.01688    | 0.02514    | 0.01954     | 0.02728     | 0.04143     | 0.05327     | 0.09671      |
>
> The peak memory consumption of DUNE is generally comparable to Dir-GAT. For small to moderate values of $T$, DUNE’s runtime is comparable to Dir-GAT, although this is dataset dependent and training DUNE can be significantly slower for large values of $T$. We would like to emphasize though that our implementation is far from optimal, so runtime improvements are certainly possible.
>
> [1] Dwivedi, V.P., Rampášek, L., Galkin, M., Parviz, A., Wolf, G., Luu, A.T. and Beaini, D., 2022. Long range graph benchmark. Advances in Neural Information Processing Systems, 35, pp.22326-22340.
>
> [2] Rossi, E., Charpentier, B., Di Giovanni, F., Frasca, F., Günnemann, S. and Bronstein, M.M., 2024, April. Edge directionality improves learning on heterophilic graphs. In Learning on graphs conference (pp. 25-1). PMLR.
>
> [3] Kiani, B., Fesser, L. and Weber, M., 2024. Unitary convolutions for learning on graphs and groups. Advances in Neural Information Processing Systems, 37, pp.136922-136961.

---

> > ### Author Response · Authors · 2025-11-27
> >
> > Dear reviewer 78Pn, we appreciate your time and careful review of our paper.
> >
> > Since we have now provided individual, point-by-point responses - including four new node-classification baselines, ablations on the long-range graph benchmark datasets, and consolidated runtime/memory results - could you please confirm that these address your earlier concerns?
> >
> > If anything remains unclear or unaddressed, we would be grateful if you could flag it so we can clarify promptly during the remainder of the discussion period. Thank you again for helping us improve the paper.

---

### Author Response · Authors · 2025-11-18
**General Response**

Thank you to all the reviewers for thoughtful and insightful feedback! We are very pleased that reviewers are excited about the novel contributions of our work. Reviewers remark that our method is **“novel and elegant”** [1L3e], appreciate our **“extensive experiments on 12 directed-graph benchmarks”** [a2Gp] which “positions unitary convolutions as a **scalable, geometry-aware approach** for deep learning on directed graphs” [kVV9]. They further acknowledged that our model “**provably avoids exponential oversmoothing** and vanishing/exploding gradient problems” [78PN] and found the paper “**well-written** and **easy to follow**” [GaaG].

We now highlight a few important points raised by reviewers that we address in our rebuttal:

- **More node-classification baseline models** to compare against were requested by reviewers 78Pn and Gaag: in response, we have added GCN and SAGE with aggregation over only incoming edges as asked for by 78Pn, and two transformer-based GNNs [Gaag]. These results further support our claims that Dune outperforms non-unitary GNNs on node classification tasks on directed graphs.
- Reviewers 78Pn and Gaag asked for **additional results** on datasets where very deep GNNs outperform shallow ones. To address this, we extended our ablations **on two long-range graph benchmark datasets**, which show that Dune performs best at extreme depths, and that its performance increases with the number of layers.
- A **detailed ablation on the role of the hyperparameter $T$** in the Taylor expansion was requested by reviewer 1L3e. We now include ablations across $T \in \{2, 4, 6, 8, 10\}$, clarifying the accuracy-efficiency trade-off and when lower $T$ is preferable.
- Finally, reviewers 78Pn and Gaag wanted to see **statistics on the runtime and peak memory consumption** of our proposed Dune model. In response, we provide detailed statistics for various values of T on three node classification datasets, as well as a comparison with two popular directed GNNs.

We believe that these additions have made the paper significantly better and more accessible and thank the reviewers again for their input. Below, we address each of the reviewers’ comments and questions individually.

---

### Meta-Review · Area_Chair_5ma2 · 2026-01-05

**Summary:**

Reviewers highlight strong theoretical guarantees and promising empirical results, with two clear advocates. The main decision-driving concerns were (i) scalability/complexity due to dense operators (exp-like propagation) and unclear runtime/memory tradeoffs, (ii) whether “very deep” training advantages are convincingly demonstrated beyond UniConv and across directed/edge-feature-rich settings, and (iii) presentation clarity (blurry figures, heavy exposition, and missing/unclear baselines). The rebuttal appears to address several technical questions (notably edge-feature integration and Taylor-order sensitivity). The remaining issues are essentially about scalability, evidence, and presentation, which are serious but seem addressable in a new submission.

**Reviewer Concerns:**

Addressed by rebuttal/discussion:

Authors responded directly to how edge features enter the framework while preserving guarantees, and indicated additional ablations on Taylor-order sensitivity (a key request).

One positive reviewer explicitly noted an increase in their score after the exchange, suggesting the rebuttal resolved enough of their concerns.

Still outstanding:

The scalability critique remains substantial: the operator can be dense and computationally heavy, and at least one reviewer was unconvinced that deep-training claims are borne out in the reported ablations.

Baseline coverage and clarity issues (e.g., including UniConv in oversmoothing comparisons, graph-transformer comparisons, and figure quality) remain unaddressed in the final version.

**Reviewer Scores:**

kVV9 (6): already indicated an increase; would likely remain at the updated level with full participation in discussion.

1L3e (4): likely 6 if the Taylor-order ablations and clarifications are incorporated cleanly into the paper.

GaaG (2): may remain a reject absent strong scalability validation.

78Pn (4): may increase modestly given improved empirical/scalability substantiation.

---

### Decision · Program_Chairs · 2026-01-26

Reject